# Generation of a CRF$_1$-Cre transgenic rat and the role of central amygdala CRF$_1$ cells in nociception and anxiety-like behavior

Marcus M Weera[1]*, Abigail E Agoglia[2], Eliza Douglass[2], Zhiying Jiang[3], Shivakumar Rajamanickam[3], Rosetta S Shackett[1], Melissa A Herman[2,4†], Nicholas J Justice[3,5†], Nicholas W Gilpin[1,6,7,8†]

[1]Department of Physiology, Louisiana State University Health Sciences Center, New Orleans, United States; [2]Department of Pharmacology, University of North Carolina, Chapel Hill, United States; [3]Institute of Molecular Medicine, University of Texas Health Sciences Center, Houston, United States; [4]Bowles Center for Alcohol Studies, University of North Carolina, Chapel Hill, United States; [5]Department of Integrative Biology and Pharmacology, McGovern Medical School at UT Health, Houston, United States; [6]Neuroscience Center of Excellence, Louisiana State University Health Sciences Center, New Orleans, United States; [7]Alcohol & Drug Abuse Center of Excellence, Louisiana State University Health Sciences Center, New Orleans, United States; [8]Southeast Louisiana VA Healthcare System (SLVHCS), New Orleans, United States

*For correspondence:
mweera@lsuhsc.edu

†These authors contributed equally to this work

**Competing interest:** The authors declare that no competing interests exist.

**Abstract** Corticotropin-releasing factor type-1 (CRF$_1$) receptors are critical to stress responses because they allow neurons to respond to CRF released in response to stress. Our understanding of the role of CRF$_1$-expressing neurons in CRF-mediated behaviors has been largely limited to mouse experiments due to the lack of genetic tools available to selectively visualize and manipulate CRF$_1^+$ cells in rats. Here, we describe the generation and validation of a transgenic CRF$_1$-Cre-$^{td}$Tomato rat. We report that *Crhr1* and *Cre* mRNA expression are highly colocalized in both the central amygdala (CeA), composed of mostly GABAergic neurons, and in the basolateral amygdala (BLA), composed of mostly glutamatergic neurons. In the CeA, membrane properties, inhibitory synaptic transmission, and responses to CRF bath application in $^{td}$Tomato$^+$ neurons are similar to those previously reported in GFP$^+$ cells in CRFR1-GFP mice. We show that stimulatory DREADD receptors can be targeted to CeA CRF$_1^+$ cells via virally delivered Cre-dependent transgenes, that transfected Cre/$^{td}$Tomato$^+$ cells are activated by clozapine-n-oxide in vitro and in vivo, and that activation of these cells in vivo increases anxiety-like and nocifensive behaviors. Outside the amygdala, we show that Cre-$^{td}$Tomato is expressed in several brain areas across the brain, and that the expression pattern of Cre-$^{td}$Tomato cells is similar to the known expression pattern of CRF$_1$ cells. Given the accuracy of expression in the CRF$_1$-Cre rat, modern genetic techniques used to investigate the anatomy, physiology, and behavioral function of CRF$_1^+$ neurons can now be performed in assays that require the use of rats as the model organism.

## Editor's evaluation

This manuscript details the generation and characterization of a new transgenic rat line presenting an exciting new tool to the field.

## Introduction

Corticotropin-releasing factor (CRF) is a 41 amino acid neuropeptide that acts on various cell populations in the brain and elicits stress-related behavioral (e.g. anxiety) and physiological (e.g. sympathomimetic) responses (*Vale et al., 1981*; *Henckens et al., 2016*). In the hypothalamic-pituitary-adrenal (HPA) axis, CRF released by neurons in the paraventricular nucleus of the hypothalamus (PVN) into the hypophyseal portal system stimulates adrenocorticotropic hormone (ACTH) release from the pituitary, which in turn stimulates release of glucocorticoids from the adrenal cortex. Glucocorticoids released into the circulatory system serve as effectors of the HPA axis by modulating various physiological (e.g. cardiovascular, respiratory, and immune) functions (*Smith and Vale, 2006*). Extrahypothalamic CRF neurons are located in many brain areas that modulate affective states and behaviors, such as the amygdala, cortex, hippocampus, midbrain, and locus coeruleus (*Peng et al., 2017*). Among these brain areas, the extended amygdala, particularly the central amygdala (CeA) and bed nucleus of stria terminalis (BNST), contain the highest densities of CRF neurons (*Peng et al., 2017*).

CRF type-1 (CRF$_1$) receptors are G$_s$- or G$_q$-coupled metabotropic receptors (*Milan-Lobo et al., 2009*; *Wanat et al., 2008*) that are highly expressed in brain areas that contain high densities of CRF fibers and/or CRF neurons, including the CeA, BNST, and medial amygdala. CRF signaling via CRF$_1$ modulates neurophysiological processes underlying stress reactivity, anxiety-related behaviors, learning and memory processes including fear acquisition and/or expression, pain signaling, and addiction-related behaviors (see reviews by *Dedic et al., 2018*; *Henckens et al., 2016*). Therefore, elucidation of the circuit-specific and cell-specific mechanisms by which CRF$_1$ signaling mediates these processes represents an important avenue of research for understanding behaviors related to stress, anxiety, fear, pain, and addiction.

The central amygdala (CeA) is a key nucleus in modulating affective states and behaviors related to stress, anxiety, fear, pain, and addiction (*Gilpin et al., 2015*). Functionally, the CeA serves as a major output nucleus of the amygdala and can be anatomically divided into lateral (CeAl) and medial (CeAm) subdivisions (*Duvarci and Pare, 2014*). In mice and rats, CRF-expressing (CRF$^+$) neurons are densely localized to the CeAl, whereas CRF$_1$-expressing (CRF$_1^+$) neurons are mostly localized to the CeAm (*Day et al., 1999*; *Jolkkonen and Pitkänen, 1998*; *Justice et al., 2008*; *Pomrenze et al., 2015*). In addition to receiving CRF input from CRF$^+$ neurons in the CeAl, the CeAm receives CRF inputs from distal brain areas such as the bed nucleus of stria terminalis (BNST) (*Dabrowska et al., 2016*) and dorsal raphe nucleus (*Commons et al., 2003*). CeA CRF$_1$ signaling plays an important role in modulating affective states and behaviors. For example, pharmacological blockade of CeA CRF$_1$ attenuates stress-induced increases in anxiety-like behavior (*Henry et al., 2006*), nociception (*Itoga et al., 2016*), and alcohol drinking (*Roberto et al., 2010*; *Weera et al., 2020*), as well as fear acquisition and expression (*Sanford et al., 2017*). Work using transgenic CRF and CRF$_1$ reporter and Cre-driver mice have shed light on circuit- and cell-specific mechanisms by which CeA CRF$^+$ and CRF$_1^+$ cells mediate affective behaviors (e.g. *Fadok et al., 2017*; *Sanford et al., 2017*). However, mice have a limited behavioral repertoire when compared to rats, making the study of more complex behaviors (e.g. drug self-administration and social interaction; *Homberg et al., 2017*) difficult.

Here, we describe the generation of a novel CRF$_1$-Cre-$^{td}$Tomato transgenic rat line. We show that *Crhr1* and *iCre* mRNA are expressed in the same neurons in the CeA and BLA using hybridization histochemistry. In the CeA, we show that $^{td}$Tomato$^+$ neurons are abundant in the CeAm, and that they are surrounded and in contact with CRF-containing puncta. We recorded membrane properties, inhibitory synaptic transmission, and spontaneous firing in CeA CRF$_1$-Cre-$^{td}$Tomato cells and show that these cells are sensitive to CRF. In addition, we show that Cre-dependent DREADD receptors can be targeted for expression by CeA CRF$_1$ cells such that DREADD stimulation of CRF$_1^+$ CeA neurons increases nociception and anxiety-like behaviors, recapitulating prior work using pharmacological strategies. Outside the amygdala, we analyzed the expression of CRF$_1$-Cre-$^{td}$Tomato cells in several brain areas across the rostrocaudal axis and show that these cells are found in brain areas known to contain CRF$_1$-expressing cells. These anatomical, electrophysiological, and functional data support the utility of this CRF$_1$-Cre-$^{td}$Tomato transgenic rat line for the study of CRF$_1$ neural circuit function, and will be an important new resource that will complement CRFR1:GFP and CRFR1:Cre mice (*Justice et al., 2008*; *Jiang et al., 2018*; *Sanford et al., 2017*), and CRF-Cre rats (*Pomrenze et al., 2015*) for the study of CRF signaling in physiology and behavior.

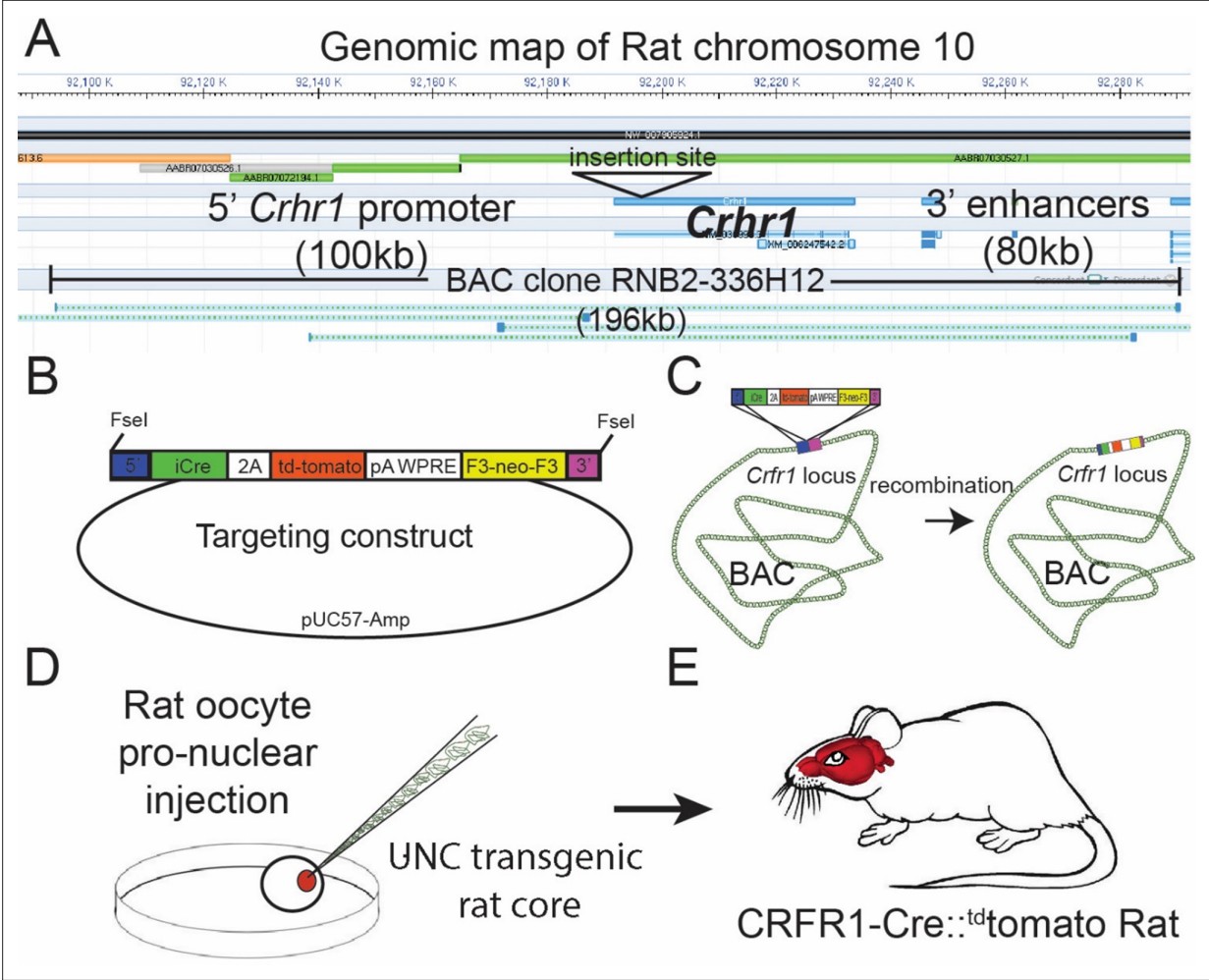

**Figure 1.** Design of the Crhr1-Cre2aTom BAC transgene. (**A**)*Crhr1* is located on chromosome 10 in the rat. A BAC clone containing 196 kb of DNA surrounding the *Crhr1* coding region includes 100 kb of upstream and 80 kb of downstream DNA, where the majority of promoter and enhancer sequences that control Crhr1 expression are located, was obtained (Riken, RNB2-336H12). There are no other sequences within this 196 kb DNA clone have been annotated as coding sequences for genes other than Crhr1. (**B**) A transgene containing 5′ (blue) and 3′ (magenta) targeting sequences, a bicistronic iCre 2 A fused tdTomato (red) sequence, 3′ polyA/WPRE stabilizing sequence, and a F3 flanked neomycin resistance sequence (yellow) was constructed then transformed into *E. coli* containing the RNB2-336H12 BAC construct. (**C**) Using recombineering techniques we isolated BAC clones in which targeted insertion of the transgene at the translation start site of *Crhr1* (ATG) was confirmed by PCR/sequencing. A single Bacterial clone containing the transgene inserted BAC was sent to the UNC transgenic facility where BAC DNA was purified and injected into single cell, fertilized rat oocytes. Two independent rat lines were recovered in which the entire BAC sequence (confirmed by PCR) was inserted into genomic DNA, of which one line displayed transgenic expression in a pattern representative of known *Crhr1* expression patterns.

## Results

### Generation of CRF$_1$-Cre-$^{td}$Tomato rats and validation of iCre ($^{Td}$Tomato) expression in CRF$_1$-expressing cells in the CeA

Design of CRF$_1$-Cre bacterial artificial chromosome (BAC) transgene

Please refer to *Figure 1* for a schematic of the BAC generation. The design of the CRF$_1$:Cre rat BAC transgene is similar to the design used to generate CRFR1:Cre BAC transgenic mice (*Jiang et al., 2018*), with the exception that a BAC clone from rat was used (clone RNB2-336H12 from Riken Rat genomic BAC library). The BAC clone used lacks any other identified protein encoding genetic sequences, reducing the possibility that other genes will be expressed from BAC when introduced into the rat genome. We obtained the RNB2-336H12 clone from the Riken Rat BAC clone collection (*STAR Consortium et al., 2008*). This BAC was inserted using recombineering techniques such that

sequences encoding *Crhr1* were replaced, beginning at the translation ATG start site, with sequences encoding a bicistronic *iCre-2A-*<sup>td</sup>*Tomato* transgene (*Figure 1C*). *iCre* is a codon-optimized Cre that produces consistent Cre activity (*Koresawa et al., 2000*). <sup>td</sup>Tomato encodes a red fluorescent protein that allows cells expressing the transgene to be visualized (*Shaner et al., 2004*). The poly A and WPRE sequences stabilize the mRNA to achieve more robust expression (*Glover et al., 2002*). To insert the iCre-p2A-tdTomato cassette into the BAC, we transformed bacterial cells that contain the BAC with a helper plasmid (Portmage-4) which contains a heatshock-inducible element that drives expression of lambda red recombinase and confers chloramphenicol resistance (*Liu et al., 2003*). These cells were then transformed with the iCre-p2A-<sup>td</sup>Tomato homology arm targeting cassette, after a 15 min heatshock. Cells were selected on kanamycin (for neoR) and colonies were screened by PCR for insertion of the cassette at the site of *Crhr1* in the BAC. BAC DNA from a single clone that contained the correct insertionwas purified, then transformed into EL250 cells, which carry an arabinose inducible flipase construct. Cultures of EL250 carrying the inserted BAC were induced to express flp recombinase by incubating in L-arabinose for 1 hr, then selected on ampicillin (the resistance of the BAC) and screened for loss of kanamycin resistance (removed by flp recombinase; *Figure 1*). Colonies were PCR screened to confirm that they contained BAC DNA containing the Cre-2A-Tom transgene inserted in the *Crhr1* locus, and lacked the f3 flanked neoR cassette. A single bacterial clonecontaining the final full-length inserted BAC construct was sent to the University of North Carolina (UNC) Transgenic Core, where BAC DNA was purified, checked for integrity by DNA laddering with EcoR1 and XbaI followed bypulsed-gel electrophoresis, and by PCR, linearized, then injected into single-cell Wistar rat oocytes.

## Generation of transgenic rats

Transgenic rats were generated by the UNC Transgenic Core by injecting single-cell rat oocytes with the modified BAC described above (*Figure 1D and E*). DNA from F1 offspring were tested for the presence of the introduced BAC transgene using PCR. Complete BAC insertion was determined using four primer sets representing unique junctions in the BAC. Two animals containing BAC insertions were further tested using this procedure and were found to contain the entire BAC sequence. Transgenic rats were first outcrossed to wildtype Wistar animals, then intercrossed to maximize the genetic similarity of transgenic offspring. Transgenic animals were generated on a Wistar backgroud because Wistar rats are commonly used in models of alcohol and substance use disorders, and other models of behavioral and physiological disorders.

## Validation of iCre (<sup>td</sup>Tomato) expression in CeA CRF$_1$ cells

The purpose of this experiment was to determine the pattern of <sup>td</sup>Tomato protein expression and *Crhr1* and *iCre* mRNA expression within the CeA. Immunohistochemical labeling of <sup>td</sup>Tomato in the amygdala showed that <sup>td</sup>Tomato$^+$ cells were located in the lateral, basolateral, central, and medial amygdala (*Figure 2A*). Within the CeA, <sup>td</sup>Tomato$^+$ cells were found to be concentrated in the CeAm, whereas the CeAl was largely devoid of <sup>td</sup>Tomato$^+$ cells (*Figure 2B*).

We used RNAscope ISH to test the hypothesis that *Crhr1* and *iCre* mRNA are highly colocalized within the CeA. <sup>td</sup>Tomato mRNA was not probed because iCre and <sup>td</sup>Tomato are expressed as a single polypeptide (iCre-2A-<sup>td</sup>Tomato) that is cleaved at the 2 A site. RNAscope ISH probing of *Crhr1* and *iCre* mRNA showed strong expression of these molecules within the CeAm (*Figure 2C–G*). Since previous work (e.g., *Justice et al., 2008*) and our data show that CRF$_1$$^+$ cells are largely localized to the CeAm, analysis of *Crhr1* and *iCre* mRNA expression was focused on this subregion. Quantification of *Crhr1*- and *iCre*-expressing cells within the CeAm showed that more than 90% of *Crhr1*-expressing cells co-express *iCre* (*Figure 2H, I*). There were no significant sex differences in the number of *Crhr1*$^+$, *Cre*$^+$, and *Crhr1*$^+$/*Cre*$^+$ +, but there was a trend for more *Crhr1*$^+$ cells in the CeAm of male rats (p = 0.07).

Previous work showed that CeA CRF$^+$ cells are concentrated in the CeAl, whereas CRF$_1$$^+$ cells are mostly located in the CeAm (*Day et al., 1999*; *Jolkkonen and Pitkänen, 1998*; *Justice et al., 2008*; *Pomrenze et al., 2015*). Immunofluorescent labeling of CRF and <sup>td</sup>Tomato protein in the CeA of CRF$_1$-Cre-<sup>td</sup>Tomato rats shows that the topography of CRF$^+$ and <sup>td</sup>Tomato$^+$ (CRF$_1$-Cre) cells in these rats is consistent with previous studies (*Figure 3*).

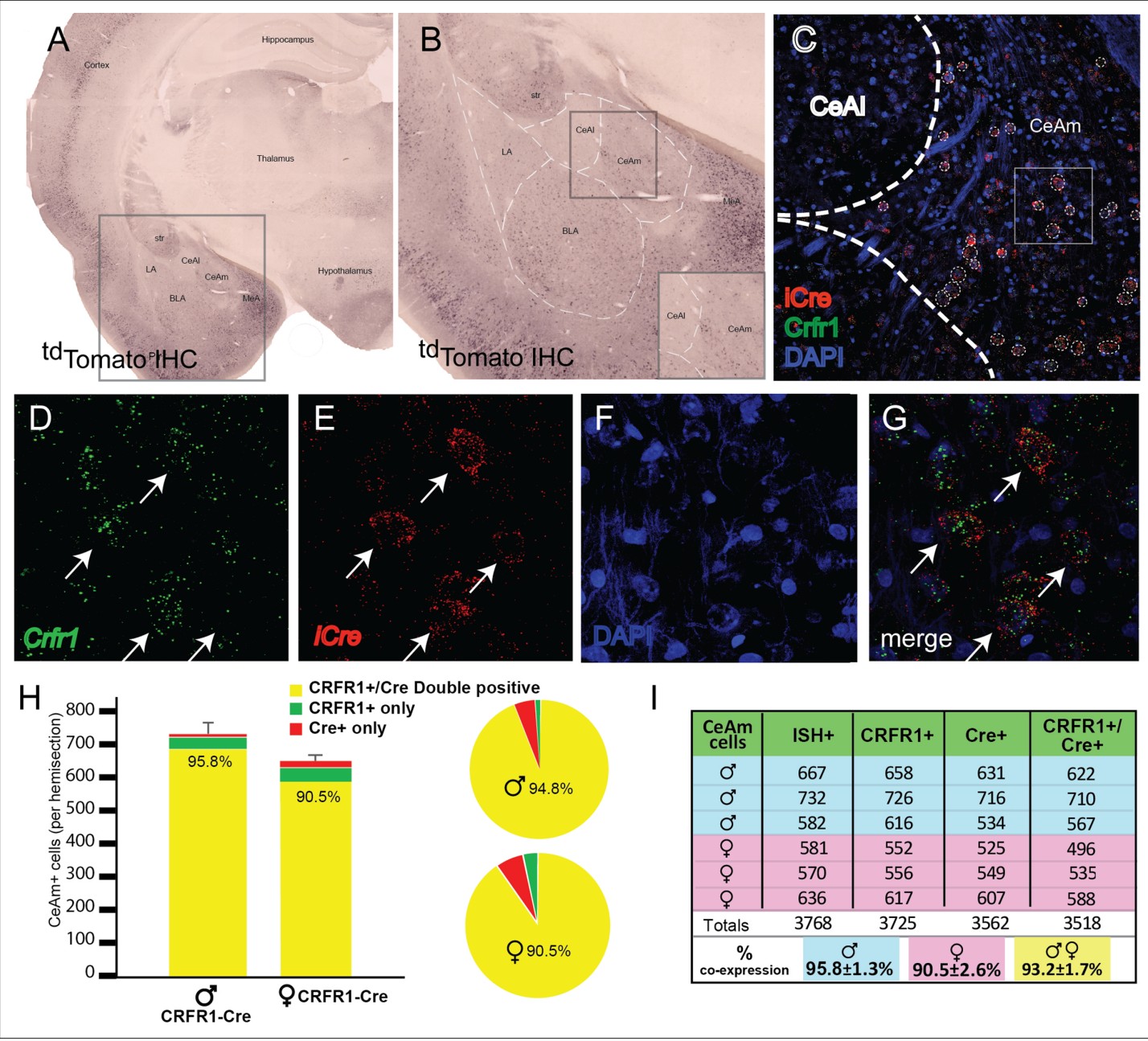

**Figure 2.** Validation of transgenic expression of iCre/<sup>td</sup>Tomato in CRF₁⁺ expressing neurons located in the medial central nucleus of the amygdala (CeAm). (**A**) A low-magnification image of a section containing the amygdala from a CRF₁-Cre rat, immunohistochemically labeled for <sup>td</sup>Tomato. Expression of the CRF₁-Cre transgene is broadly very similar to previous reports of CRF₁ expression in both rat and mouse. (**B**) Within the boxed region of panel A, higher magnification reveals CRF₁⁺ cells in the lateral amygdala (LA), basolateral amygdala (BLA), medial portion central amygdala (CeAm), and medial amygdala (MeA). The lack of significant labeling in the CeAl is consistent with reports using both in situ hybridization and transgenic reporters to detect CRF₁ expression. (**C**) Micrograph of the CeA from the region boxed in panel B allows visualization of mRNA encoding *iCre* (red) and *Crhr1* (green) along with nuclei stained with DAPI (blue). (**D–G**) Higher magnification images of the boxed region in (**C**) allows visualization of mRNA for *Crhr1* (D, green), and *iCre* (E, red), with nuclei visualized by DAPI staining (**F**). (**G**) Merged images reveals that many CeAm neurons that are positive for *Crhr1* mRNA are also positive for *iCre* mRNA (arrows point to double positive neurons). Quantification of coincidence of in situ hybridization for both *Crhr1* and *iCre* mRNAs demonstrates that >90% of *Crhr1*-positive cells are also positive for *iCre* in the CeAm (n = 3). (**H**) Graphical representation of quantification of coincident labeling, or (**I**) a table of the precise counts from each of three male and three female CRF₁-Cre transgenic animals. We observed greater than 90% of neurons positive for both *Crhr1* and *iCre* in the CeAm.

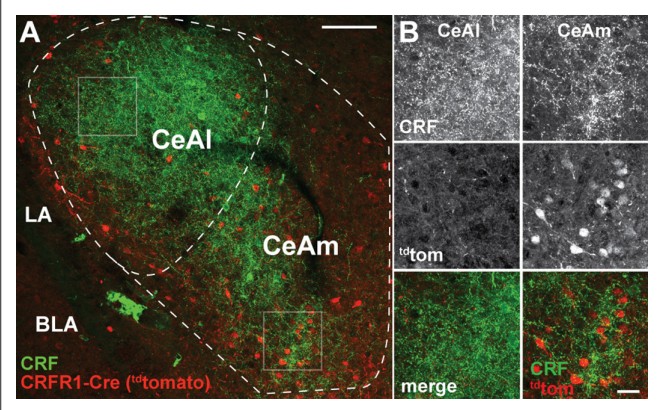

**Figure 3.** CRF$_1$-driven expression of Cre/$^{td}$Tomato in the CeA. (**A**) Visualization of CRF using immunofluorescent labeling (green) in a rat carrying the CRF$_1$-Cre2aTom transgene (red) reveals minimal cellular expression of CRF$_1$ in the lateral central nucleus of the amygdala (CeAl) where CRF is highly abundant. This discrepancy in CRF localization compared to CRF$_1$ expression is consistent with previous reports of CRF$_1$ expression in both rat and mouse. In contrast to the CeAl, the medial central nucleus of the amygdala (CeAm) contains many CRF$_1^+$ neurons (reported by the CRF$_1$-Cre2aTom transgene), in contact with puncta positive for CRF peptide. (**B**) High-resolution images from the boxed regions of CeAl and CeAm in panel A. CRF staining is dense in both the CeAl and CeAm (top panels); however, cellular expression of the CRF$_1$-Cre2aTom transgene is low in the CeAl, while many neurons in the CeAm are positive for CRF$_1$ expression (middle panels). Merged images (lower panels) display the coincident staining of CRF$_1^+$neurons with CRF puncta in the CeAm, suggesting that stress driven CRF release directly signals to CRF$_1^+$ neurons in the CeAm to modulate neural excitability to influence the output of CeAm neurons. LA – lateral amygdala, BLA – basolateral amygdala.

## Electrophysiological characterization of CeA CRF$_1$-Cre-$^{td}$Tomato neurons
### Membrane properties and inhibitory transmission

$^{td}$Tomato$^+$ CRF$_1^+$ neurons were identified and differentiated from unlabeled CeA neurons using fluorescent optics and brief ( < 2 s) episcopic illumination in slices from adult male and female CRF$_1$-Cre-$^{td}$Tomato rats. Consistent with our immunohistochemical studies (*Figure 3*), the majority of CRF$_1^+$ neurons were observed in the medial subnucleus of the CeA (CeAm) and this region was targeted for recordings. Passive membrane properties were determined during online voltage-clamp recordings using a 10 mV pulse delivered after break-in and stabilization. The resting membrane potential was determined online after breaking into the cell using the zero current (I = 0) recording configuration. No differences were observed between male and female membrane properties including membrane capacitance, membrane resistance, decay time constant, or resting membrane potential (*Figure 4A*). CRF$_1^+$ CeA neurons were then placed in current clamp configuration and a depolarizing step protocol was conducted to allow cell-typing based on previously described firing properties (*Chieng et al., 2006*; *Herman and Roberto, 2016*). The majority of CRF$_1^+$ CeA neurons were of the low-threshold bursting type (*Figure 4B*). Voltage-clamp recordings of pharmacologically-isolated GABA$_A$ receptor-mediated spontaneous inhibitory postsynaptic currents (sIPSCs) revealed that CRF$_1^+$ neurons are under a significant amount of phasic inhibition (*Figure 4C*) with no significant sex differences in sIPSC frequency (*Figure 4D*, **left**) or sIPSC amplitude (*Figure 4D*, **right**).These data indicate that male and female CRF$_1^+$ CeA neurons have similar basal membrane properties and are under similar levels of basal inhibitory transmission.

### CRF sensitivity

Spontaneous firing activity was recorded in CRF$_1^+$ CeA neurons from male and female CRF$_1$-Cre-$^{td}$Tomato rats using the cell-attached configuration. After a stable baseline period of regular firing was established, CRF (200 nM) was focally applied and the firing activity was recorded for a sustained application period of 7–12 min. CRF$_1^+$ CeA neurons from male rats had an average baseline firing rate of 2.1 ± 0.5 Hz and focal application of CRF significantly increased the firing activity to 3.4 ± 0.7 Hz

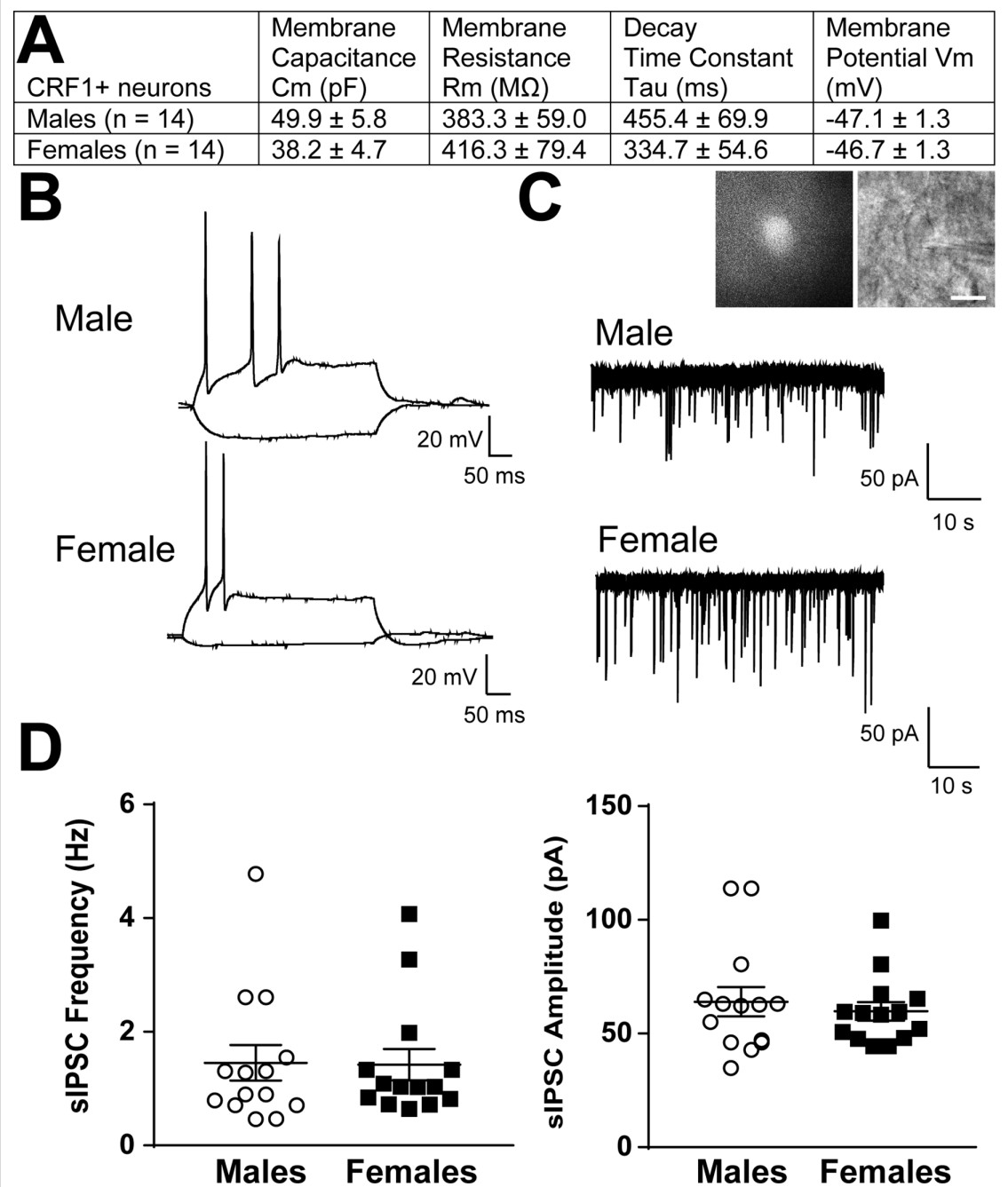

| CRF1+ neurons | Membrane Capacitance Cm (pF) | Membrane Resistance Rm (MΩ) | Decay Time Constant Tau (ms) | Membrane Potential Vm (mV) |
|---|---|---|---|---|
| Males (n = 14) | 49.9 ± 5.8 | 383.3 ± 59.0 | 455.4 ± 69.9 | -47.1 ± 1.3 |
| Females (n = 14) | 38.2 ± 4.7 | 416.3 ± 79.4 | 334.7 ± 54.6 | -46.7 ± 1.3 |

**Figure 4.** Basal membrane properties and inhibitory synaptic transmission in CeAm CRF$_1$-Cre-$^{td}$Tomato neurons. (**A**) Basal membrane properties (Membrane Capacitance, Cm; Membrane Resistance, Rm; Decay Constant, Tau; Membrane Potential, Vm) from male and female CRF$_1^+$ CeAm neurons. (**B**) Representative current-evoked spiking properties from male (top) and female (bottom) CRF$_1^+$ CeAm neurons. (**C**) Basal spontaneous inhibitory postsynaptic currents (sIPSCs) from male (top) and female (bottom) CRF$_1^+$ CeAm neurons (right). *Inset*: representative fluorescent (left) and infrared differential interference contract (IR-DIC, right) image of a CRF$_1^+$ CeAm neuron targeted for recording. Scale bar = 20 μm. (**D**) Average sIPSC frequency (left) and sIPSC amplitude (right) from male and female CRF$_1^+$ CeAm neurons. Raw data are available in Source Data File 1.

(t = 3.5, p = 0.011; *Figure 5A and C*). CRF$_1^+$ CeA neurons from female rats had an average baseline firing rate of 0.8 ± 0.2 Hz and focal application of CRF significantly increased the firing activity to 1.3 ± 0.3 Hz (t = 3.1, p = 0.016 by paired t-test; *Figure 5B and D*). When firing activity was normalized to baseline values, CRF application significantly increased firing in CRF$_1^+$ CeA neurons from male rats to 192.3% ± 25.6% of Control (t = 3.6, p = 0.009; *Figure 5E*) and significantly increased firing in CRF$_1^+$

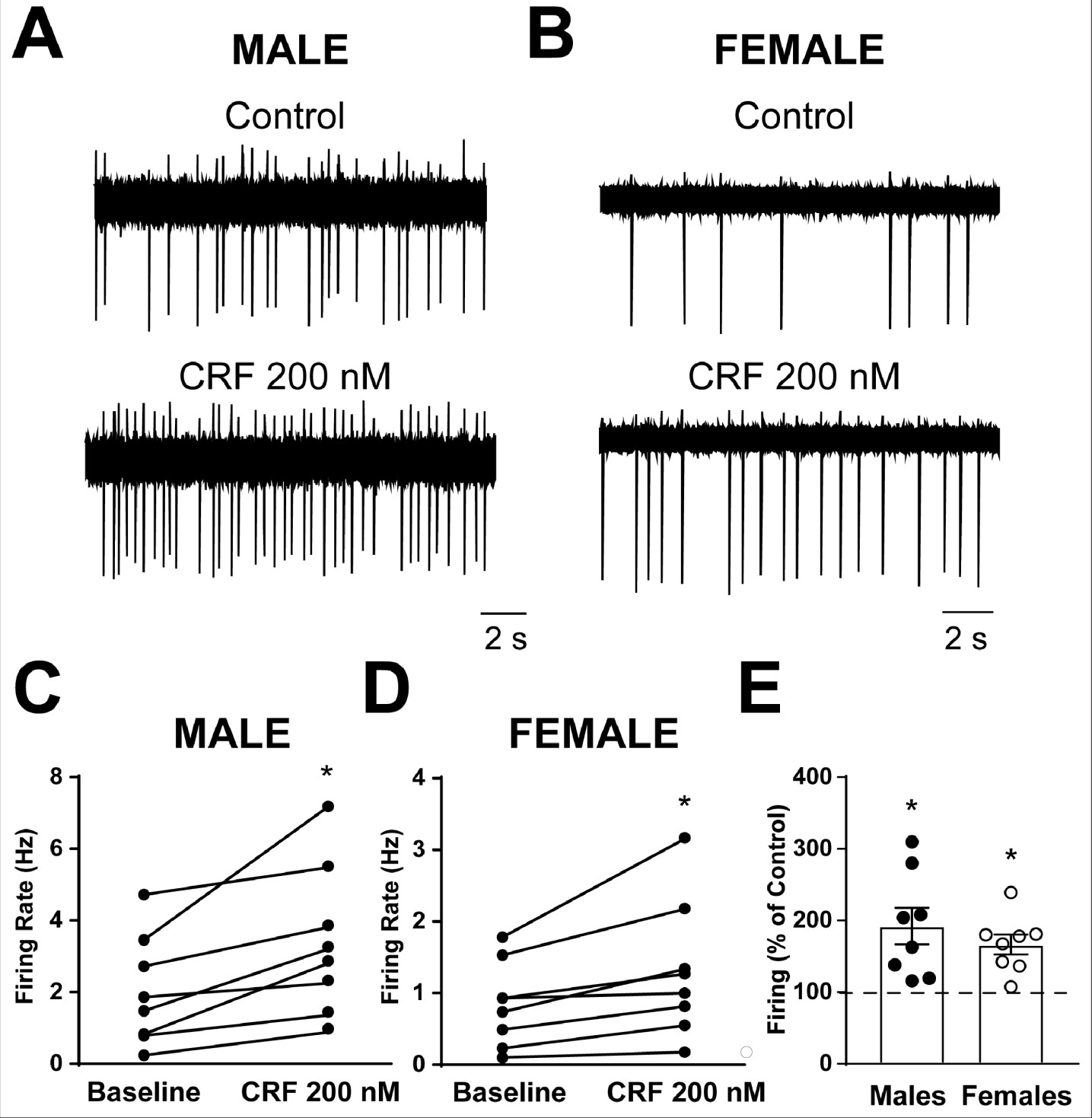

**Figure 5.** Spontaneous firing activity and CRF sensitivity of CeAm CRF$_1$-Cre-$^{td}$Tomato neurons. (**A**) Representative cell-attached recording of spontaneous firing activity in a CRF$_1^+$ CeAm neuron from a male CRF$_1$-Cre-$^{td}$Tomato rat before and during CRF (200 nM) application. (**B**) Representative cell-attached recording of spontaneous firing activity in a CRF$_1^+$ CeAm neuron from a female CRF$_1$-Cre-$^{td}$Tomato rat before and during CRF (200 nM) application. (**C**) Summary of changes in spontaneous firing activity with CRF application in CRF$_1^+$ CeAm neurons from male CRF$_1$-Cre-$^{td}$Tomato rats.*p < 0.05 by paired t-test. (**D**) Summary of changes in spontaneous firing activity with CRF application in CRF$_1^+$ CeAm neurons from female CRF$_1$-Cre-$^{td}$Tomato rats.*p < 0.05 by paired t-test. (**E**) Normalized change in firing activity in CRF$_1^+$ CeAm neurons from male and female CRF$_1$-Cre-$^{td}$Tomato rats.*p < 0.05 by one-sample t-test. Raw data are available in Source Data File 1.

CeA neurons from female rats to 166.5% ± 13.9% of Control (t = 4.8, p = 0.002; *Figure 5E*) with no significant difference in the change in firing in response to CRF between male and female $CRF_1^+$ neurons.

Targeting of Cre-dependent Gq-DREADDs to the CeA increases $c\text{-}Fos^+$ $CRF_1$-Cre-$^{td}$Tomato cells and CeA $CRF_1$-Cre-$^{td}$Tomato cell activity following CNO treatment *c-Fos immunohistochemistry:* Four weeks after $CRF_1$-Cre-$^{td}$Tomato male and female rats were given intra-CeA microinjections of AAV8-hSyn-DIO-HA-hM3D(Gq)-IRES-mCitrine (Active Virus) or AAV5-hSyn-DIO-EGFP (Control Virus), rats were given an injection (i.p.) of CNO (4 mg/kg) and were sacrificed 90 min later. Brain sections containing the CeA were processed for c-Fos immunohistochemistry and the number of $c\text{-}Fos^+$ $^{td}$Tomato cells were quantified. An overwhelming majority of $^{td}$Tomato cells were located in the CeAm as shown above (*Figure 3*). Therefore, quantification of c-Fos and $^{td}$Tomato cells was focused on this subregion. We found that rats in the active virus group had a higher percentage of $c\text{-}Fos^+$ $^{td}$Tomato cells than rats in the control virus group (*t* = 7.1, p < 0.001; *Figure 6A and B*).

## Slice electrophysiology

Coronal sections containing the CeA were prepared from male and female $CRF_1$-Cre-$^{td}$Tomato rats > 4 weeks after rats were given intra-CeA microinjections of AAV8-hSyn-DIO-HA-hM3D(Gq)-IRES-mCitrine. $^{td}$Tomato$^+$ and mCitrine$^+$ neurons were identified by brief episcopic illumination using fluorescent optics and positively-identified neurons were targeted for recording in whole-cell current clamp configuration to measure changes in resting membrane potential and spontaneous firing. CNO (10 μM) significantly increased membrane potential and number of action potentials in both male (t = 4.3, p = 0.002; t = 2.6, p = 0.030, respectively; *Figure 6C and D*) and in female $CRF_1^+$ mCitrine$^+$ neurons in the CeA (t = 3.2, p = 0.016; t = 3.6, p = 0.006, respectively; *Figure 6E and F*), suggesting that hM3D(Gq) receptor expression was functional and could be stimulated by CNO application with no sex differences in expression or agonist sensitivity.

## Chemogenetic stimulation of CeA $CRF_1$-Cre-$^{td}$Tomato cells increases mechanical nociception and anxiety-like behaviors

Rats were given intra-CeA microinjections of a viral vector for Cre-dependent expression of Gq-DREADD receptors (AAV8-hSyn-DIO-HA-hM3D(Gq)-IRES-mCitrine) or control fluorophore (AAV5-hSyn-DIO-EGFP) (*Figure 7A*). Behavioral procedures began ≥4 weeks later (*Figure 7B*).

## Nociception

In the Von Frey test of mechanical nociception, CNO treatment decreased paw withdrawal thresholds in rats that have hM3D(Gq) expression targeted to CeA $CRF_1^+$ cells [repeated measures ANOVA; test x treatment interaction ($F_{1,13}$ = 14.0, p = 0.002)]. There was a significant effect of test within the CNO group only [($F_{1,7}$ = 38.0, p < 0.001)], suggesting that chemogenetic stimulation of CeA $CRF_1^+$ cells increases mechanical sensitivity. CNO treatment had no effect on paw withdrawal thresholds in rats that received the control EGFP fluorophore (*Figure 7C*). In the Hargreaves test of thermal nociception, CNO did not affect paw withdrawal latencies in either the hM3D(Gq) or control EGFP groups (*Figure 7D*).

## Anxiety-like behaviors

In the EPM test, in the hM3D(Gq) group, rats that were given CNO treatment had lower open arms time compared to rats that were given vehicle treatment (t-test; *t* = 2.6, p = 0.022), suggesting that chemogenetic stimulation of CeA $CRF_1^+$ cells increases anxiety-like behavior on the EPM. Control EGFP virus rats did not show differences in open arms times after CNO treatment (*Figure 7E*). One rat in the hM3D(Gq) group that was given vehicle injection fell off the maze >3 times and was therefore excluded from analysis. In the OF field test, hM3D(Gq) rats that were given CNO treatment spent less time in the center of the arena compared to rats that were given vehicle treatment (t-test; *t* = 3.3, p = 0.006), suggesting that chemogenetic stimulation of CeA $CRF_1^+$ cells increases anxiety-like behavior in

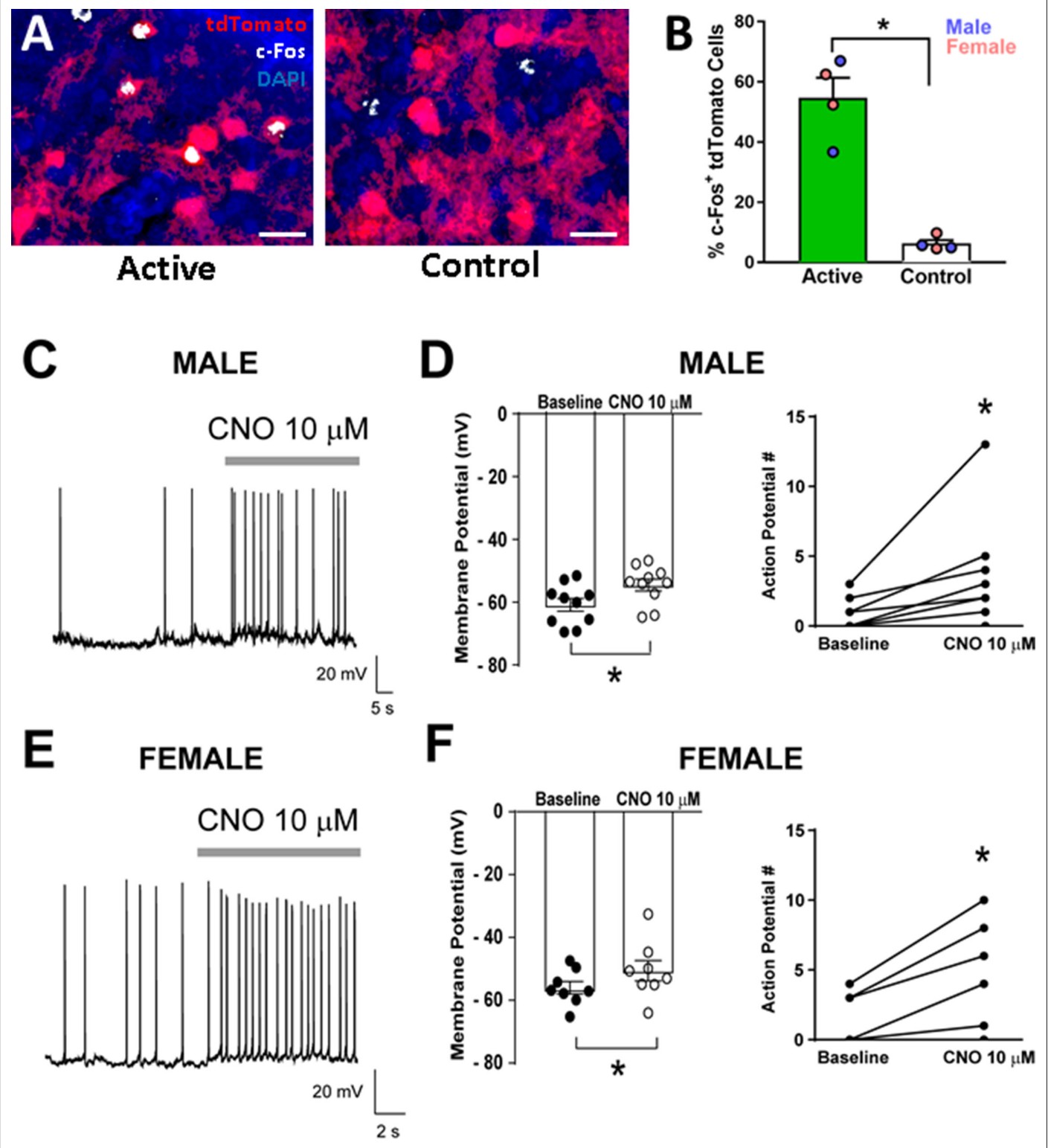

**Figure 6.** Validation of DREADD expression and function in CeA CRF$_1$-Cre-$^{td}$Tomato neurons. (**A**) Representative images of CRF$_1$-Cre-$^{td}$Tomato cells (red) and c-Fos immunostaining (white) in CeAm of rats that were given intra-CeA microinjections of AAV8-hSyn-DIO-HA-hM3D(Gq)-IRES-mCitrine (active virus) or AAV5-hSyn-DIO-EGFP (control virus). Scale bar: 50 µm. (**B**) CNO treatment 90 min before sacrifice increased the percentage of c-Fos$^+$ $^{td}$Tomato cells in CeAm of rats that were given active virus compared to rats that were given control virus microinjections. *p < 0.05. (**C**) Representative whole-cell current clamp recording of membrane potential and firing activity in a CRF$_1^+$ CeAm neuron from a male CRF$_1$-Cre-$^{td}$Tomato rat before and during CNO (10 µM) application. (**D**) Summary of the change in membrane potential (left) and action potentials (right) in male CRF$_1^+$ CeAm neurons after CNO

*Figure 6 continued on next page*

*Figure 6 continued*

application. *p < 0.05 by paired t-test. (**E**) Representative whole-cell current clamp recording of membrane potential and firing activity in a $CRF_1^+$ CeAm neuron from a female $CRF_1$-Cre-tdTomato rat before and during CNO (10 µM) application. (**D**) Summary of the change in membrane potential (left) and action potentials (right) in female $CRF_1^+$ CeAm neurons after CNO application. *p < 0.05 by paired t-test. Raw data are available in Source Data File 1.

the OF. EGFP control rats did not show differences in time spent in the center of the arena after CNO treatment (***Figure 7F***). In the LD test, CNO treatment did not produce differences in time spent in the light box in either hM3D(Gq) or EGFP control groups (***Figure 7G***). There were also no differences in latency to enter the light box (data not shown; 20.14 ± 5.63 s, 29.38 ± 10.42 s, 23.00 ± 3.94 s, and 16.17 ± 4.48 s, respectively, for rats in hM3D(Gq) – Veh, hM3D(Gq) – CNO, EGFP – Veh, and EGFP Virus – CNO groups).

Given that most cells in the CeA are GABAergic, we next tested if *Crhr1* and *iCre* mRNA are also highly colocalized in the BLA, which contains a high density of glutamatergic neurons that express $CRF_1$ receptors. RNAscope ISH of brain sections containing BLA show dense *Crhr1* and *iCre* mRNA expression in BLA and surrounding areas (***Figure 8A***). Quantification of *Crhr1* and *iCre* mRNA in the BLA showed that 98.5% of *Crhr1*-expressing cells co-express *iCre* (***Figure 8B***, top), and that 99.4% of *iCre*-expressing cells co-express *Crhr1* (***Figure 8B***, bottom).

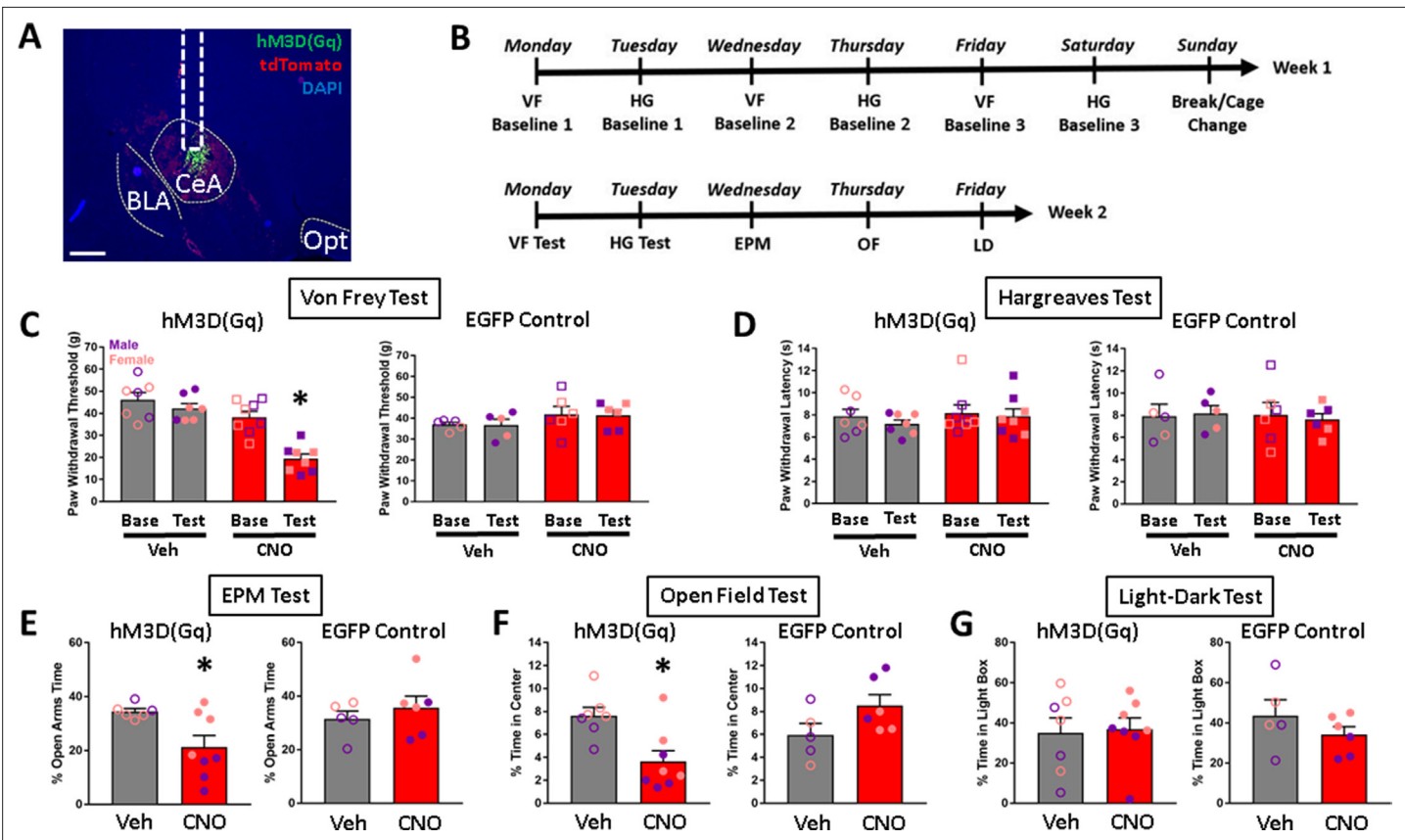

**Figure 7.** Effects of chemogenetic stimulation of CeA $CRF_1$-Cre-tdTomato neurons on nociception and anxiety-like behaviors. (**A**) Representative image of AAV8-hSyn-DIO-HA-hM3D(Gq)-IRES-mCitrine expression (green) in the CeA. Scale bar: 500 µm. BLA: basolateral amygdala, Opt: optic tract. (**B**) Timeline of experimental procedures. (**C**) CNO treatment decreased paw withdrawal thresholds in the Von Frey test of mechanical nociception in rats that were given intra-CeA hM3D(Gq) virus microinjections. There were no effects of treatment on paw withdrawal thresholds in the EGFP control group. (**D**) CNO treatment had no effects on paw withdrawal latencies in either the hM3D(Gq) or EGFP groups in the Hargreaves test of thermal nociception. (**E**) CNO treatment decreased the percent time spent in open arms in the EPM test in the hM3D(Gq) group, but had no effect in the EGFP group. (**F**) CNO treatment decreased the percent time spent in the center of the arena in the OF test in the hM3D(Gq) group, but had no effect in the EGFP group. (**G**) CNO treatment had no effect on percent time spent in the light box in the LD test. *p < 0.05. Raw data are available in Source Data File 2. *iCre* and *Crhr1* mRNA are highly co-expressed in BLA.

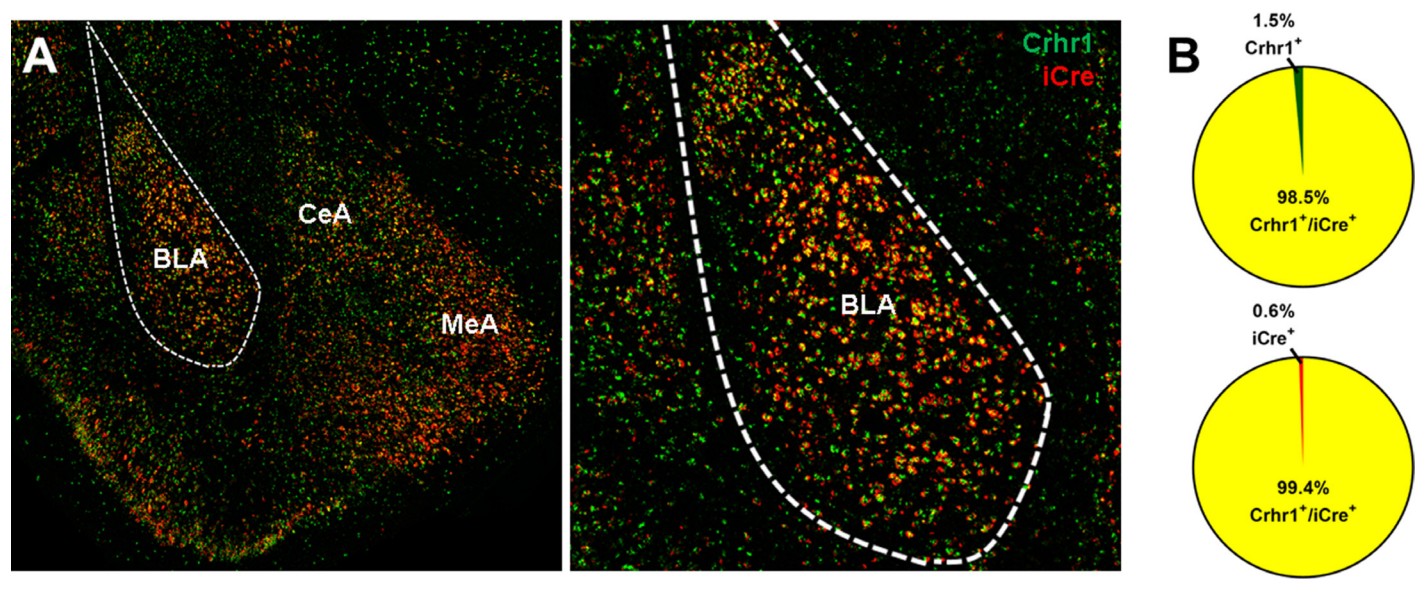

**Figure 8.** *Crhr1* and *iCre* mRNA expression in BLA of CRF$_1$-Cre-$^{td}$Tomato rats. (**A**) *Crhr1* (green) and *iCre* (red) mRNA are highly co-expressed in BLA and surrounding areas. (**B**) Within the BLA, 98.5% of *Crhr1*$^+$ + co-express *iCre* (top), and 99.4% of *iCre*$^+$ + co-express *Crhr1* (bottom).

## Cre-$^{td}$Tomato cells are found throughout the brain in areas known to contain CRF$_1$ cells

The purpose of this experiment is to examine if the Cre-$^{td}$Tomato transgene is expressed in brain areas outside of the amygdala in a pattern that is consistent with known CRF$_1$ expression patterns. Fluorescence imaging of coronal sections across the rostrocaudal axis of CRF$_1$-Cre-$^{td}$Tomato rat brains show that Cre-$^{td}$Tomato cells are found in brain areas known to contain CRF$_1$ cells, including the prelimbic and infralimbic cortex, BNST, piriform cortex (PIR), PVN, lateral hypothalamus (LHA), paraventricular thalamus (PVT), hippocampus, ventral tegmental area (VTA), periaqueductal grey (PAG), and dorsal raphe (DR) (***Figure 9***). The density of Cre-$^{td}$Tomato cells in these brain areas were surveyed and compared to published reports of *Crhr1* mRNA expression patterns in the wildtype rat brain (***Van Pett et al., 2000***) and of CRF$_1$-GFP cell densities in CRF$_1$-GFP mouse brain (***Justice et al., 2008***; ***Table 1***). We found that a majority of cells in the BNST and PIR, that a substantial number of cells in the PrL, IL, PVT, VTA, PAG, and DR, and that some cells in the ventral lateral septum (LSv), PVN, LHA, and hippocampus are $^{td}$Tomato$^+$. The lateral (LHb) and medial habenula (MHb) were largely devoid of tdTomato$^+$ cells, but a small number of $^{td}$Tomato cells were found in the ventricular zone of the MHb (***Figure 9Q***). The locus coeruleus (LC) was devoid of $^{td}$Tomato cells, but a small number of $^{td}$Tomato puncta was detected (***Figure 9X***). These $^{td}$Tomato expression patterns are mostly consistent with Crhr1 mRNA expression patterns in wildtype rats, but some species-specific differences are seen when compared to GFP expression patterns in CRF$_1$-GFP mice. Please see ***Table 1***.

## Discussion

We generated a new transgenic CRF$_1$-Cre-$^{td}$Tomato rat line to allow genetic manipulation and visualization of neurons that express CRF$_1$ receptors in the rat brain. We report that, within the CeA of CRF$_1$-Cre-$^{td}$Tomato rats, CRF$_1$-Cre-$^{td}$Tomato cells are located in the medial subdivision (CeAm), consistent with previous reports of *Crhr1* expression in rats (***Potter et al., 1994***; ***Van Pett et al., 2000***; ***Day et al., 1999***) and mice (***Van Pett et al., 2000***; ***Justice et al., 2008***), and that there is strong concordance ( > 90%) between *Crhr1* and *iCre* mRNA expression in the CeAm of male and female CRF$_1$-Cre-$^{td}$Tomato rats. We also characterized the basal membrane properties, inhibitory synaptic transmission, and validated the CRF sensitivity of $^{td}$Tomato-expressing CeA CRF$_1$$^+$ cells in male and female rats. In addition, we showed that stimulatory DREADD receptors [hM3D(Gq)] can be targeted to CeA CRF$_1$-Cre-$^{td}$Tomato cells using a Cre-dependent expression strategy, that systemic CNO treatment

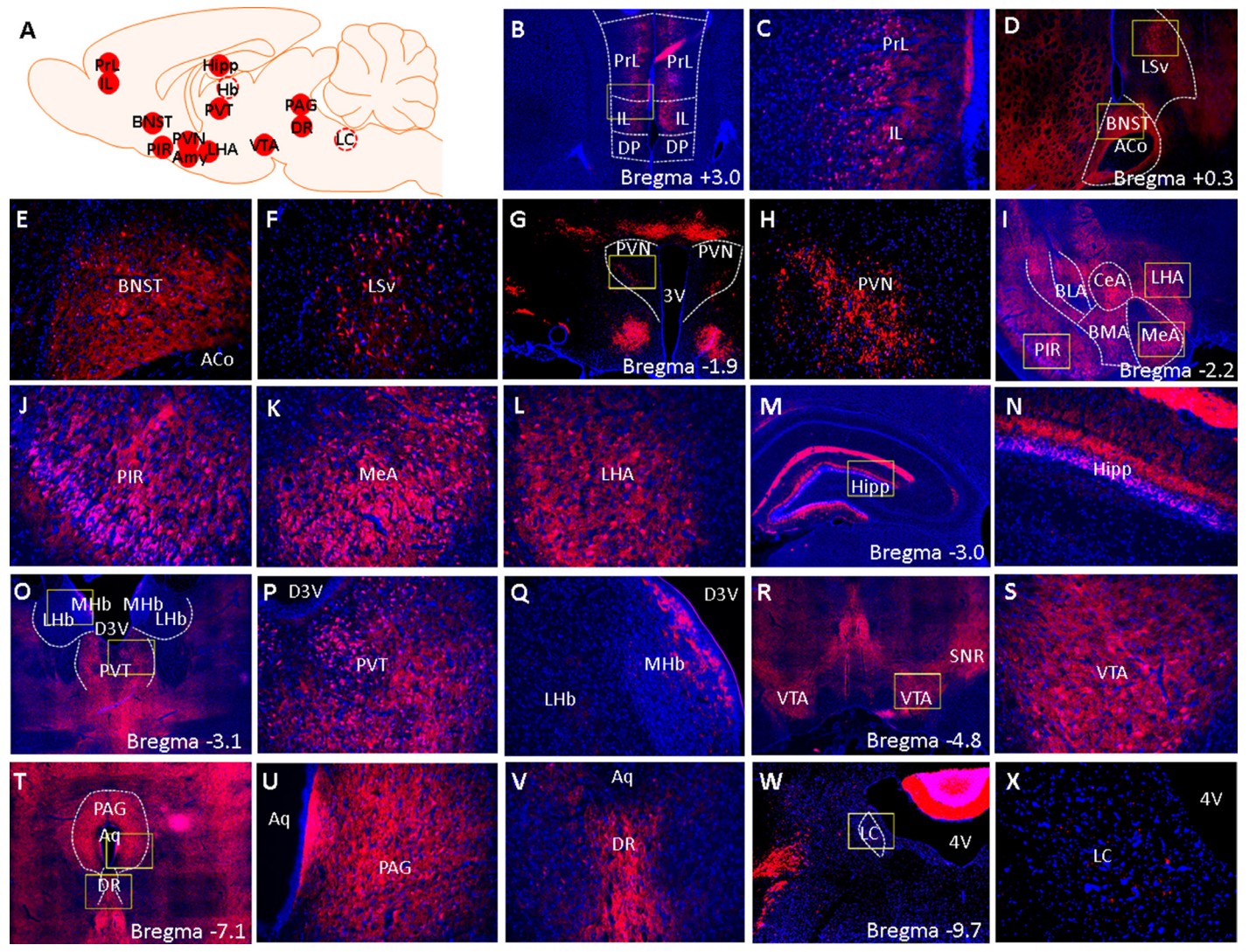

**Figure 9.** Fluorescent images of brainwide tdTomato expression. (**A**) A schematic summary of brain areas that were surveyed for tdTomato expression. Solid red circles indicate brain areas that contain tdTomato cells and dashed circles indicate brain areas that were devoid of tdTomato cells. (**B–X**) Low (4 x) and high-magnification (×20) images of tdTomato (red) and DAPI (blue) fluorescent signals. Yellow boxes demarcate areas from which high-magnification (×20) images were acquired. (**B**) 4 x image of the PrL, IL, and surrounding landmarks. (**C**) 20 x image of PrL and IL. (**D**) 4 x of BNST and LSv. (**E**) 20 x image of BNST and (**F**) LSv. (**G**) 4 x and (**H**) 20 x image of PVN. (**I**) 4 x image of PIR, MeA, and LHA. (**J**) 20 x image of PIR, (**K**) MeA, and (**L**) LHA. (**M**) 4 x and (**N**) 20 x image of hippocampus. (**O**) 4 x image PVT, LHB, and MHb. (**P**) 20 x image of PVT and (**Q**) LHb/MHb. (**R**) 4 x and (**S**) 20 x image of VTA. (**T**) 4 x image of PAG and DR. (**U**) 20 x image of PAG and (**V**) DR. (**W**) 4 x and (**X**) 20 x image of LC. 3 V: 3rd ventricle, 4 V: 4th ventricle, D3V: dorsal 3rd ventricle, Aq: cerebral aqueduct.

induces c-Fos in hM3D(Gq)-transduced $CRF_1^+$ cells in CeA, and that CNO induces membrane depolarization and spontaneous firing activity in hM3D(Gq)-transduced $CRF_1^+$ cells in CeA. We showed that hM3D(Gq)-mediated stimulation of CeA $CRF_1^+$ cells increases anxiety-like behavior, as measured by EPM and open field tests, as well as mechanical nociception as measured by the Von Frey test. Finally, we report that *Crhr1* and *iCre* mRNA expression are highly colocalized in the BLA, and that Cre-tdTomato is expressed in several brain areas across the rostrocaudal axis of the rat brain in expected patterns. Collectively, this work provides cellular, electrophysiological, and behavioral data demonstrating the validity and reliability of a new transgenic rat model for the identification and selective manipulation of $CRF_1^+$ neurons in the CeA.

$CRF_1$-Cre-tdTomato rats were generated by modifying a BAC clone derived from F344 rats, that was injected into oocytes derived from Wistar rats. Because the originating strain of the BAC DNA differs

**Table 1.** Comparison of CRF$_1$-Cre-tdTomato cell densities with Crhr1 mRNA expression levels in rats and CRF$_1$-GFP cell densities in mice in several areas across the rostrocaudal axis of the brain.
+: Some cells are positive, ++: A substantial number of cells are positive, +++: Most cells are positive.

| Brain area | Density of tdTomato cells in CRF1-Cre-tdTomato rats | Expression level of Crhr1 mRNA in wildtype rats (**Van Pett et al., 2000**) | Density of GFP cells in CRF$_1$-GFP Mice (**Justice et al., 2008**) |
|---|---|---|---|
| PrL | ++ | ++ | ++ |
| IL | ++ | ++ | ++ |
| BNST | +++ | +++ | +/++ |
| LSv | + | + | - |
| PVN | + | + | ++ |
| PIR | +++ | +++ | ++/+++ |
| MeA | ++ | + | +/++ |
| LHA | + | + | ++ |
| Hipp | + (subiculum, CA3)/+++ (CA1) | ++ (CA1, CA3, subiculum) | + (subiculum, CA3)/+++ (CA1) |
| PVT | ++ | ++ | - |
| LHb | - | - | ++ |
| MHb | + | - | - |
| VTA | ++ | ++ | ++ |
| PAG | ++ | + | + |
| DR | ++ | + | ++ |
| LC | - | - | +/- |

from the strain used for transgenesis, DNA sequence differences between F344 and Wistar rats in the 200 + kb region surrounding the *Crhr1* genomic locus may impact expression of iCre-2A-$^{td}$Tomato in the CRF$_1$-Cre transgenic rat. Moreover, because transgenic BAC DNA insertion was not directed, position effect, copy number, and fragmentation of the BAC insertion may possibly alter expression of reporter genes. These sources of variability were minimized by the expertise of the UNC Transgenic Core Facility, where BAC DNA was prepared, cleaved at a single site to produce full-length single stranded BAC DNA, then injected into oocytes. Two transgenic founders were isolated that contained unique DNA sequences near the *Crhr1* genomic locus in the BAC, as well as DNA sequence from both terminal ends of the injected BAC, indicating insertion of full-length BAC DNA. Of these two rats, one displayed expression of $^{td}$Tomato in a pattern reflecting *Crhr1* expression patterns reported in rats (*Potter et al., 1994*; *Van Pett et al., 2000*), as well as transgenic expression of reporter genes reported in similarly designed BAC transgenic mice (*Justice et al., 2008*; *Jiang et al., 2018*; *Hunt et al., 2018*). The genomic locus and copy number of transgenic BAC insertion have not been further characterized. However, the high degree of alignment of Cre-$^{td}$Tomato expression with known CRF$_1$ expression patterns broadly throughout the brain suggests that position, copy number, or fragmentation did not substantially alter expression of the transgene. In addition, the behavioral and physiological phenotypes of CRF$_1$-Cre-$^{td}$Tomato rats reported here closely resemble those seen in wild-type Wistar rats (e.g., *Itoga et al., 2016*; *Albrechet-Souza et al., 2020*; see below), suggesting that BAC insertion did not produce observable anomalous phenotypes.

The intrinsic properties of CRF$_1$$^+$ ($^{td}$Tomato$^+$) CeA neurons in CRF$_1$-Cre-$^{td}$Tomato rats are similar to those reported in wild-type rat CeA neurons (*Herman and Roberto, 2016*), suggesting that the transgene did not significantly impact the overall health or activity of CRF$_1$$^+$ CeA neurons. Previous studies have employed a CRFR1:GFP transgenic mouse model to examine CRF$_1$$^+$ CeA neurons in local CeA microcircuitry (*Herman et al., 2013*; *Herman et al., 2016*). Although prior work was conducted in mouse and some species differences would be expected, the electrophysiological properties of CRF$_1$$^+$ neurons in the CeA are relatively consistent between the mouse and rat transgenic models. Recent work specifically examining sex differences in CRF$_1$$^+$ neurons from CRFR1:GFP mice reported similar

intrinsic membrane properties, cell-typing, and baseline inhibitory transmission as reported here and noted no sex differences in basal properties between male and female $CRF_1^+$ neurons in the CeA (*Agoglia et al., 2020*), consistent with our current findings. In contrast to previous work, however, we found no sex differences in CRF sensitivity of $CRF_1^+$ cells in CeA. Although CRF was previously found to increase firing in $CRF_1^+$ CeA neurons in both male and female mice, $CRF_1^+$ CeA neurons from male mice displayed a significantly greater increase in firing in response to CRF application (*Agoglia et al., 2020*). This discrepancy may be driven by differences in sampling size (although the current study was not sufficiently powered to detect sex differences), high variability in male baseline firing rates observed here, sex differences in baseline firing rates observed here, or it may reflect a species differ-ence in sex-dependent sensitivity to CRF. Additional studies are required for a more comprehensive examination of sex- and/or species-specific differences in CRF-stimulated $CRFR1^+$ neuronal activity in distinct brain regions. $CRF_1^+$ neurons in the CeA have also previously been implicated in the neuro-plastic changes associated with acute and chronic ethanol exposure in mice (*Herman et al., 2013*; *Herman et al., 2016*), and future work will determine if this is also the case in rats.

To test the feasibility of using Cre-dependent stimulatory DREADDs to interrogate the role of CeA $CRF_1^+$ cells in behavior, we targeted Gq-coupled DREADD receptors [DIO-hM3D(Gq)] to CeA $CRF_1$-Cre cells and first tested the effects of CeA $CRF_1^+$ cell stimulation on nociception. We showed that DREADD stimulation of CeA $CRF_1^+$ cells increases mechanical sensitivity as measured by the Von Frey test, but not thermal sensitivity measured by the Hargreaves test. Numerous studies have implicated CeA $CRF$-$CRF_1$ signaling in nociception. For instance, pain induced by carrageenan injec-tion into the knee joint produces hyperactivity of CeA neurons in rats, a phenomenon that is $CRF_1$ dependent (*Ji and Neugebauer, 2007*). Conversely, latency for hind limb withdrawal reflex induced by knee joint pressure application is decreased (reflecting hyperalgesia) by CRF infusion into the CeA, a phenomenon that is blocked by co-infusion of a $CRF_1$ antagonist (*Ji et al., 2013*). With regard to mechanical sensitivity, our finding that DREADD activation of CeA $CRF_1$ neurons produces mechanical hypersensitivity extends previous work showing that ablation or inhibition of CeA CRF neurons blocks neuropathic pain-induced mechanical hypersensitivity, as measured by the Von Frey test (*Andreoli et al., 2017*). Various studies have also previously reported that CeA $CRF_1$ antagonism attenuates mechanical hypersensitivity produced by chronic drug or alcohol exposure (e.g. *Cohen et al., 2015*; *Edwards et al., 2012*). Collectively, this prior work suggests that activation of CeA $CRF$-$CRF_1$ signaling by chronic injuries or insults facilitates a hyperalgesic state, which was recapitulated in the current study via chemogenetic stimulation of CeA $CRF_1$ cells. It should be noted that chemogenetic stim-ulation of CeA $CRF^+$ cells has been shown to potentiate stress-induced analgesia (*Andreoli et al., 2017*), although this effect may be due to CRF release outside the CeA (e.g. locus coeruleus) or due to changes in GABA release from $CRF^+$ cells. Within the CeA, pharmacological studies suggest that pro- and anti-nociceptive effects of CRF signaling may be mediated by $CRF_1$ and $CRF_2$ receptors, respectively (*Ji and Neugebauer, 2007*; *Ji and Neugebauer, 2008*), but further studies are required to delineate the precise mechanisms by which CeA CRF signaling supports hyperalgesia and anal-gesia. The $CRF_1$-Cre-$^{td}$Tomato rat described here represents a useful tool for these studies.

Our lab has shown that predator odor stress-induced hyperalgesia, as measured by the Hargreaves test, is mediated by increased $CRF$-$CRF_1$ signaling in the CeA (*Itoga et al., 2016*). Contrary to our hypothesis, we did not observe any effect of DREADD activation of CeA $CRF_1$ cells on thermal sensi-tivity. It is not clear why $CRF_1$-Cre rats exhibited mechanical hypersensitivity but not thermal hyper-algesia in this study. It is possible that specific CeA cell populations are involved in specific types of nociceptive processing, or it is possible that the engagement/recruitment of CeA $CRF_1^+$ cells in mediating specific types of nociception depends on the animal's history or affective state. Using the $CRF_1$-Cre-$^{td}$Tomato rat line reported here, future work will elucidate the role of specific $CRF_1^+$ circuits in mechanical and thermal sensitivity (which may or may not be partially overlapping) under basal and challenged conditions such as neuropathic or inflammatory pain, stress exposure and/or withdrawal from chronic exposure to drugs or alcohol.

CeA $CRF$-$CRF_1$ signaling generally promotes anxiogenic responses, particularly under challenged conditions such as stress or withdrawal from chronic exposure to drugs of abuse. For instance, in mice, intra-CeA infusion of a $CRF_1$ antagonist attenuates anxiety-like behavior, as measured by open field and light-dark box tests, following immobilization stress but not under basal conditions (*Henry et al., 2006*). Similarly, blockade of $CRF$-$CRF_1$ signaling in the CeA attenuates alcohol withdrawal-induced

anxiety in rats, as measured by the EPM test (*Rassnick et al., 1993*). Exposure to stressors (*Merlo Pich et al., 1995*) and alcohol withdrawal (*Zorrilla et al., 2001*) both increase extracellular CRF levels in the CeA. Messing and colleagues used CRF-Cre rats to show that CeA CRF cell activation (using DREADDs) increases anxiety-like behavior and that this effect is blocked by CeA CRF knockdown using RNA interference (*Pomrenze et al., 2019*). Collectively, these studies show that increased CRF-$CRF_1$ signaling in CeA, whether produced by stress exposure, drug withdrawal, or the use of viral genetic tools, supports anxiogenesis. Here, we showed that chemogenetic stimulation of CeA $CRF_1^+$ cells increases anxiety-like behavior, as measured by EPM and open field tests. Interestingly, we did not observe an effect of chemogenetic stimulation of CeA $CRF_1^+$ cells on light-dark box measures, including time spent in the light vs. dark boxes, latency to enter light box, and number of crosses between the two compartments. Previous studies reported that the light-dark box test may be more suited for detecting anxiolytic rather than anxiogenic effects (*Crawley and Davis, 1982*), and that animals' responses in this test is affected by the intensity of illumination in the animal's regular housing room (*File et al., 2004*). Future studies will test if Cre-dependent expression of inhibitory chemogenetic or optogenetic strategies can be applied to inhibit CeA $CRF_1$ cells, and if inhibition of CeA $CRF_1$ cells reduces anxiety-like behavior at baseline or after stress exposure.

Although the CeA is generally not thought to exhibit strong sexual dimorphism, sex differences have been reported for CRF and $CRF_1$ properties in the CeA. For example, female rats in proestrus have higher levels of CeA *Crf* mRNA than male rats and footshock stress induces greater CeA *Crf* expression in female proestrus than in male rats (*Iwasaki-Sekino et al., 2009*). However, using autoradiography, Cooke and colleagues reported no sex differences in $CRF_1$ receptor binding in the CeA of rats (*Weathington et al., 2014*). Using a transgenic CRFR1:GFP mouse line, the CeA of male animals was shown to contain more $CRF_1^+$ cells than female animals (*Agoglia et al., 2020*). In the current study, although underpowered to detect sex effects, we found that male $CRF_1$-Cre-tdTomato rats tend to have more $CRF1^+$ cells within the CeA than female rats, as detected via RNAscope in situ hybridization. Previous work using CRFR1:GFP mice (*Agoglia et al., 2020*) and our current work using $CRF_1$-Cre-tdTomato rats revealed no sex differences in basal membrane properties and inhibitory synaptic transmission in CeA $CRF_1^+$ neurons. Our previous work in CRFR1:GFP mice showed that, while CRF application increased firing in $CRF_1^+$ CeA neurons from both female and male mice, larger increases were observed in $CRF_1^+$ neurons from male mice, an effect that we did not observe in rats potentially due to differences in variability or sex differences in firing rate.

The overwhelming majority of published studies examining the role of CeA $CRF_1$ signaling in nocifensive and anxiety-like behaviors has employed only male subjects. Here, we used both sexes in tests of nociception and anxiety-like behavior and we did not detect any sex differences. However, we observed that the anxiogenic effect of CeA $CRF_1^+$ cell activation in the EPM test was driven by stronger effects in male subjects. One interpretation is that CeA $CRF_1^+$ cell activation affects specific components of anxiety-related behavior, that these components are aligned with specific 'anxiety-like behavior' phenotypes in male versus female rats that are more or less detectable by these various tests. In support of this idea, a meta-analysis of studies of anxiety-like behavior in rats and mice revealed a discordance in results between the EPM and open field tests (*Mohammad et al., 2016*). However, due to the small number of studies testing female animals, the role of sex in this discordance is still unknown. Within the EPM literature, studies have shown that the EPM test is more reliable for detecting anxiogenic effects in male rather than female rodents (e.g. *Scholl et al., 2019*). Collectively, these findings demonstrate the importance of behavioral assay selection and of using both male and female subjects.

In summary, we present a novel $CRF_1$-Cre-tdTomato rat line that allows for the visualization and manipulation of $CRF_1$-expressing cells. The $CRF_1$-Cre rat can be used to study the role of distinct $CRF_1^+$ cell populations and circuits in physiology and behavior. In addition, the expression of the fluorescent reporter tdTomato in $CRF_1$-expressing cells will allow for high resolution, detailed analysis of the expression pattern of $CRF_1$ that has not been possible due to the lack of $CRF_1$-specific antibodies.

# Materials and methods

**Key resources table**

| Reagent type (species) or resource | Designation | Source or reference | Identifiers | Additional information |
|---|---|---|---|---|
| Gene (*Rattus norvegicus*) | BAC containing the *Crfr1* genomic locus | Riken Gene Engineering Division | RNB2-336H12 | |
| Transfected construct (*Rattus norvegicus*) | *iCre-2a-tdTomato* BAC transgene | This paper | | See Results, Design of CRF$_1$-Cre BAC; Contact Justice Lab |
| Genetic reagent (*Rattus norvegicus*) | CRF1-Cre-$^{td}$Tomato rat | This paper | | See Results, Generation of Transgenic Rats; Contact Gilpin Lab |
| Recombinant DNA reagent | AAV8-hSyn-DIO-HA-hM3D(Gq)-IRES-mCitrine | Addgene | Cat# 50454-AAV8 | |
| Recombinant DNA reagent | AAV5-hSyn-DIO-EGFP | Addgene | Cat# 50457-AAV5 | |
| Antibody | Anti-c-Fos (Rabbit polyclonal) | Abcam | Cat# ab190289, RRID:AB_2737414 | (1:1000) |
| Antibody | Anti-HA-Tag (Rabbit monoclonal) | Cell Signaling | Cat# 3724, RRID:AB_1549585 | (1:250) |
| Antibody | Anti-RFP (Rabbit monoclonal) | Abcam | Cat# ab34771, RRID:AB_777699 | (1:500) |
| Antibody | Anti-CRF (Rabbit monoclonal) | The Salk Institute | Rc-68 | (1:2000) |
| Commercial assay or kit | RNAscope Multiplex Fluorescent Kit v2 | ACD Bio | *iCre* and *Crfr1* probes | |
| Commercial assay or kit | TSA Detection Kit | Akoya Biosciences | Cat# NEL701A001KT | |
| Commercial assay or kit | Prolong Gold Antifade Reagent with DAPI | Invitrogen | Cat# P36935 | |
| Chemical compound, drug | Clozapine-n-oxide | NIH Drug Supply Program | | |
| Chemical compound, drug | Corticotropin-releasing factor | Tocris | Cat# 1,607 | |
| Software, algorithm | Mini Analysis | Synaptosoft Inc. | RRID:SCR_002184 | |
| Software, algorithm | Clampfit 10.6 | Molecular Devices | | |
| Software, algorithm | Prism 7.0 | GraphPad | RRID:SCR_002798 | |
| Software, algorithm | SPSS 25 | IBM SPSS | RRID:SCR_019096 | |

## Subjects

Adult male and female CRF$_1$-Cre-$^{td}$Tomato rats were used in all experiments. Rats bred from the original founder F1 line were group-housed in humidity- and temperature-controlled (22 °C) vivaria at UNC, LSUHSC, or UTHSC on a reverse 12 hr light/dark cycle (lights off at 7:00 or 8:00 AM) and had ad libitum access to food and water. All behavioral procedures occurred in the dark phase of the light-dark cycle. Sample sizes were estimated using published work from our labs. In situ hybridization and immunohistochemistry experiments were performed in a single pass. Behavioral experiments were performed in two independent, fully counterbalanced replicates. Slice electrophysiology experiments were conducted in a single pass, such that one experiment was performed in each slice and each experimental group contained neurons from a minimum of three rats. All procedures were approved by the Institutional Animal Care and Use Committee of the respective institutions at which procedures occurred (LSUHSC IACUC Protocol #3749; UNC IACUC Protocol #19–190; UTHSC IACUC Protocol #21–075) and were in accordance with National Institutes of Health guidelines.

## Stereotaxic surgeries

Rats were anesthetized with isoflurane (2 min at 4% for induction, 1%–3% for maintenance) and mounted into a stereotaxic frame (Kopf Instruments) for all stereotaxic surgeries. The following coordinates (from bregma) were used for bilateral intra-CeA microinjections: –2.5 mm posterior, ± 4.0 mm lateral, and –8.4 mm ventral for male rats and –2.2 mm posterior, ± 4.0 mm lateral, and –8.2 mm ventral for female rats. Viral vectors for Cre-dependent expression of Gq-DREADDs or control (see below) were injected into each side of the CeA at a volume of 0.5 µL over 5 min and injectors were

left in place for an additional 2 min. Viral titers were between 1.0–1.5 x $10^{13}$ GC/mL. At the end of surgeries, rats were monitored to ensure recovery from anesthesia and were given a minimum of 4 weeks to recover before the start of procedures. Rats were treated with the analgesic flunixin (2.5 mg/kg, s.c.) and, in some rats, the antibiotic cefazolin (20 mg/kg, i.m.) before the start of surgeries and once the following day.

### Immunohistochemistry

Rats were deeply anesthetized with isoflurane and were transcardially perfused with ice-cold phosphate buffered saline (PBS) followed by 4% paraformaldehyde (PFA). Brains were extracted and post-fixed in 4% PFA for 24 hr (at 4 °C), cryoprotected in 20% sucrose for 48–72 hr (at 4 °C), and frozen in 2-methylbutane on dry ice. Coronal sections were collected using a cryostat and stored in 0.1% sodium azide in PBS at 4 °C until further processing.

### c-Fos immunofluorescent labeling

Sections (40 μm) containing the CeA were washed 3 × 10 min in PBS and incubated in blocking buffer (2.5% normal goat serum with 0.3% Triton X-100) for 2 hr at RT. Subsequently, sections were incubated in rabbit anti-c-Fos polyclonal antibody (1:1000 in blocking buffer; catalog no. 190289, Abcam, Cambridge, United Kingdom) for 48 hr at 4 °C. Sections were then washed 3 × 10 min in PBS and incubated in goat anti-rabbit Alexa Fluor 647 (1:500 in blocking buffer; catalog no. A32733, Invitrogen, Carlsbad, CA) for 2 hr at RT. After 3 × 10 min washes in PBS, sections were mounted on microscope slides and coverslipped with Prolong Gold Antifade Reagent with DAPI (Invitrogen, catalog no. P36935). Sections were imaged using a Keyence (Osaka, Japan) BZ-X800 fluorescent microscope at ×20 magnification and Fos$^+$ cells were quantified manually by a blinded experimenter. Four sections representative of the CeA anterior-posterior axis (~bregma –1.8—2.8) from each animal were used for analysis.

### HA-tag immunofluorescent labeling

Sections (40 μm) containing the CeA were washed 3 × 10 min in PBS and incubated in 3% hydrogen peroxide for 5 min. Sections were then washed 3 × 10 min in PBS and incubated in a blocking buffer containing 1% (w/v) bovine serum albumin and 0.3% Triton X-100 in PBS for 1 h at room temperature (RT). Then, sections were incubated in a rabbit anti-HA monoclonal antibody (1:250 in blocking buffer; catalog no. 3724, Cell Signaling, Danvers, MA) for 48 hr at 4 °C. Sections were then washed for 10 min in TNT buffer (0.1 M Tris base in saline with 0.3% Triton X-100), incubated for 30 min in 0.5% (w/v) Tyramide Signal Amplification (TSA) blocking reagent in 0.1 M Tris base, and incubated for 30 min in ImmPRESS horseradish peroxidase horse anti-rabbit antibody (catalog no. MP-7401, Vector Laboratories, Burlingame, CA) at RT. Following 4 × 5 min washes in TNT buffer, sections were incubated in fluorescein TSA reagent (1:50 in TSA amplification diluent) for 10 min at RT. TSA blocking reagent, fluorescein TSA reagent, and TSA amplification diluent are part of the TSA detection kit (catalog no. NEL701A001KT, Akoya Biosciences, Marlborough, MA). Sections were washed 3 × 10 min in TNT buffer, mounted on microscope slides, and coverslipped with Prolong Gold Antifade Reagent with DAPI (Invitrogen, catalog no. P36935). Sections were imaged using a Keyence BZ-X800 fluorescent microscope at ×2 and ×20 magnification.

### $^{td}$Tomato DAB immunostaining

Because $^{td}$Tomato signal degrades over time, to create a permanent set of slides for anatomical analysis, we performed immunohistochemistry to label $^{td}$Tomato protein permanently. Briefly, fixed, free-floating sections (30 μm) were incubated overnight in monoclonal rabbit anti-RFP, biotin-tagged antibody (1:500; Abcam, catalog no. ab34771). Sections were then washed 3 x with PBS and incubated in streptavidin-conjugated peroxidase (DAB-elite kit) for 1 hr. After incubation with streptavidin, sections were washed 2 x in PBS, then 2 x in 0.1 M NaOAc (pH 6.0), then stained in a solution of 0.1 M NaOAC (pH 6.0) containing nickel ammonium sulfate (3%) and 5 μl of 3% $H_2O_2$. Sections were

stained for up to 10 min, then washed 2 x in NaOAc (pH 6.0), then in PBS, before being mounted on gelatin coated slides, dehydrated, and coverslipped in DPX. Bright-field images were acquired using a Cytation 5 imager (BioTek Instruments, Winooski, VT). <sup>td</sup>*Tomato and CRF immunofluorescent labeling*: Tissue processing procedures were similar to the immunofluorescent procedures described above. Sections through the amygdala were incubated in antibodies against CRF (rabbit anti-CRF, # rc-68, 1:2000, The Salk Institute) and goat-anti-RFP (1:1000; Rockland, catalog no. 200-101-379). Primary antibodies were detected by secondary anti-rabbit antibody conjugated with Alexa Fluor 488 and anti-goat antibody conjugated with Alexa Fluor 555 (Invitrogen), resulting in <sup>td</sup>Tomato protein being visible as red fluorescence and CRF peptide visible as green fluorescence. High-magnification images of the CeA were taken using a Leica (Wetzlar, Germany) Sp5 confocal microscope.

## In situ hybridization

All solutions were prepared with DEPC treated water and all tools and surfaces were wiped with RNAzap followed by DEPC treated water. Adult Crhr1:Cre -<sup>td</sup>Tomato rats (3 males and 3 females) were deeply anesthetized with Avertin (2,2,2,-tribromoethanol, 1.25% solution, 0.2 ml/10 g BW, IP), then transcardially perfused with PBS followed by 4% PFA in PBS. Brains were removed, fixed in 4% PFA at 4 °C overnight, then equilibrated in 30% sucrose, sectioned into six series of sections (30 µM, coronal sections) on a frozen sliding microtome (SM 2000R, Leica), and stored in PBS at 4 °C. Brain slices were mounted onto glass slides, dried, and went through in situ hybridization (ISH) using a RNAscope Multiplex Fluorescent kit v2 (ACDbio, Newark, CA) following the manufacturer's protocol.

## Slice electrophysiology

Following rapid decapitation, brains were extracted and sectioned as previously described (*Herman and Roberto, 2016*). Briefly, brains were placed in ice-cold high sucrose solution containing (in mM): sucrose 206.0; KCl 2.5; CaCl2 0.5; MgCl2 7.0; NaH2PO4 1.2; NaHCO3 26; glucose 5.0; HEPES 5. Coronal sections (300 µm) were prepared on a vibrating microtome (Leica VT1000S, Leica Microsystem) and placed in an incubation chamber with oxygenated (95% O2/5% CO2) artificial cerebrospinal fluid (aCSF) containing (in mM): NaCl 120; KCl 2.5; EGTA 5; CaCl2 2.0 MgCl2 1.0; NaH2PO4 1.2; NaHCO3 26; Glucose 1.75; HEPES 5. Slices were incubated for 30 min at 37 °C, followed by a 30 min acclimation at room temperature. Patch pipettes (3–6 MΩ; King Precision Glass Inc, Claremont, CA) were filled with an internal solution containing (in mM): potassium chloride (KCl) 145; EGTA 5; MgCl2 5; HEPES 10; Na-ATP 2; Na-GTP 0.2 (for whole cell voltage-clamp recordings) or containing potassium gluconate 145; EGTA 5; MgCl2 5; HEPES 10; Na-ATP 2; Na-GTP 0.2 (for whole cell current clamp experimental recordings). Data acquisition was performed with a Multiclamp 700B amplifier (Molecular Devices, San Jose, CA), low-pass filtered at 2–5 kHz, coupled to a digitizer (Digidata 1,550B; Molecular Devices) and stored on a PC using pClamp 10 software (Molecular Devices). Whole-cell voltage-clamp recordings were performed at $V_{hold}$ = –60 mV. Cell-attached recordings were performed with no holding parameters (0 mV). All recordings were performed at room temperature. Series resistance was continuously monitored and cells with series resistance >15 MΩ were excluded from analysis. Properties of sIPSCs were analyzed and visually confirmed using a semi-automated threshold detection program (Minianalysis). The frequency of firing discharge was evaluated and visually confirmed using threshold-based event detection analysis in Clampfit 10.2 (Molecular Devices). Experimental drugs were applied by bath application or y tube for focal application. Analysis was performed on recordings containing >60 events or that encompassed a period of 2–5 min.

## Drugs

Clozapine-n-oxide (CNO, NIH Drug Supply Program) was dissolved in 5% DMSO (v/v in saline). CRF was purchased from Tocris Bioscience (Bristol, United Kingdom), dissolved in stock solutions in ultrapure water, and diluted to final experimental concentration in aCSF.

## *Crhr1* and *iCre* expression in CeA

The purpose of this experiment was to determine the pattern of <sup>td</sup>Tomato protein expression and *Crhr1* and *iCre* mRNA expression within the CeA. Coronal brain sections containing the CeA were processed for immunohistochemical DAB labeling of <sup>td</sup>Tomato protein or RNAscope ISH for labeling *Crhr1* and *iCre* mRNA, as described above. A separate set of brain sections containing CeA were processed for

immunofluorescent labeling of $^{td}$Tomato and CRF to map the expression pattern of these proteins in the CeA of Crhr1-Cre-$^{td}$Tomato rats. Images were captured using a confocal microscope (model TCS SP5, Leica) and processed with Fiji ImageJ. Coronal sections containing the CeA were identified by neuroanatomical landmarks with reference to a rat brain atlas and captured at ×20 magnification at one single focal plane (1 μm). For analysis of *Crhr1* and *iCre* RNAscope images, punctate signals from each channel were quantified separately following the manufacturer's guideline (ACDbio SOP45-006). Quantification was performed by an experimenter blinded to experimental groups. Based on pilot studies, cells that had more than 3 puncta were considered positive. At least 3 sections representative of the anterior-posterior axis of the CeA were analyzed in each animal.

## Membrane properties, inhibitory synaptic transmission, and CRF sensitivity of CeA CRF$_1$-Cre-$^{td}$Tomato cells

The purpose of this experiment was to characterize intrinsic properties, inhibitory synaptic transmission, and CRF sensitivity in CRF$_1^+$ neurons in the CeA. These neurons were identified using fluorescent optics and brief ( < 2 s) episcopic illumination. All labeled neurons were photographed, recorded, and saved. Intrinsic membrane properties were determined in voltage clamp configuration ($V_{hold}$ = –60 mV) using pClamp 10 Clampex software. Current clamp recordings were performed to determine current-voltage (I-V) changes and the firing type of each neuron. Voltage clamp recordings of pharmacologically-isolated GABA$_A$ receptor-mediated spontaneous inhibitory postsynaptic currents (sIPSCs) were performed with bath application of the glutamate receptor blockers 6,7-dinitroquinoxaline-2,3-dione (DNQX, 20 μM) and DL-2-amino-5-phosphonovalerate (AP-5, 50 μM) and the GABA$_B$ receptor antagonist GCP55845A (1 μM). Cell-attached recordings were made in close juxtacellular (i.e., membrane intact) cell-attached configuration and only cells with regular spontaneous firing were included in analysis. After a stable baseline period, CRF (200 nM) was applied for a period of 7–10 min and changes in firing were measured and compared to baseline. Experiments were performed in individual slices to ensure that drug application was never repeated in the same slice.

## Functional validation of Cre-dependent expression of Gq-DREADD receptors in CeA CRF$_1$-Cre -$^{td}$Tomato cells

The purpose of this experiment was to test if CeA CRF$_1$-Cre-$^{td}$Tomato cells can be activated using chemogenetics via Cre-mediated targeting of Gq-DREADD receptors [hM3D(Gq)] to these cells.

### c-Fos validation

To test if Cre-dependent expression of Gq-DREADD receptors can stimulate CeA CRF$_1$-Cre-$^{td}$Tomato cells, CRF$_1$-Cre-$^{td}$Tomato rats were given bilateral microinjections of AAV8-hSyn-DIO-HA-hM3D(Gq)-IRES-mCitrine (50454-AAV8, Addgene, Watertown, MA) or a control virus (AAV5-hSyn-DIO-EGFP; Addgene, 50457-AAV5) targeting the CeA. Four weeks later, rats were given a systemic CNO injection (4 mg/kg, i.p.) and sacrificed 90 min later. Brain sections (4 sections/rat x 4 rats/group) were processed for c-Fos immunohistochemistry and the percentage of c-Fos$^+$ $^{td}$Tomato$^+$ cells in the CeA was calculated. Cell counts were performed by an experimenter blinded to experimental groups.

### Electrophysiological validation

To functionally validate Cre-dependent expression of Gq-DREADD receptors in CeA CRF$_1$-Cre -$^{td}$Tomato cells, CRF$_1$-Cre-$^{td}$Tomato rats were given bilateral microinjections as described above. After a minimum of 4 weeks, brain slices of CeA were prepared as described above. CeA neuronal expression of mCitrine and tdTomato were confirmed by fluorescent optics and neurons were targeted for electrophysiological recording. After a stable baseline period, CNO (10 μM) was applied and changes in membrane potential and action potential firing were measured and compared to baseline.

# Effects of chemogenetic stimulation of CeA CRF$_1$-Cre -$^{td}$Tomato cells on nociception and anxiety-like behavior

The purpose of these experiments was to test the effects of chemogenetic stimulation of CeA CRF$_1$ cells on nociception and anxiety-like behavior. Rats were given intra-CeA microinjections of AAV8-hSyn-DIO-HA-hM3D(Gq)-IRES-mCitrine (Addgene, 50454-AAV8; Active Virus) or AAV5-hSyn-DIO-EGFP (Addgene, 50457-AAV5; Control Virus) and were given 4 weeks for recovery and viral expression (*Figure 7A*). Please refer to *Figure 7B* for a timeline schematic of this experiment. All rats were habituated to handling before the start of behavioral procedures. On behavioral procedure days, rats were given at least 30 min to acclimate to the procedure room.

## Nociception

Mechanical and thermal nociception were measured using the Von Frey (*Pahng et al., 2017*) and Hargreaves (*Avegno et al., 2018*) assays, respectively, as previously described. Briefly, the Von Frey apparatus consists of clear chambers placed on top of a mesh floor. To measure sensitivity to mechanical nociception, each hind paw was perpendicularly stimulated with a Von Frey filament (Electronic Von Frey 38450, Ugo Basile, Gemonio, Italy) calibrated to measure the amount of force applied using the up-down method and the force (g) threshold required to elicit a paw withdrawal response was recorded. Force thresholds were measured twice for each hind paw in alternating fashion, with at least 1 min between measurements, and an average threshold was calculated for each animal. The Hargreaves apparatus consists of clear chambers placed on top of a glass pane suspended above a tabletop. To measure sensitivity to thermal nociception, each hind paw was stimulated by a halogen light heat source (Model 309 Hargreaves Apparatus, IITC Life Sciences, Woodland Hills, CA) and latency (s) for hind paw withdrawal was measured. Withdrawal latencies were measured twice for each hind paw in alternating fashion, with at least 1 min between measurements, and an average withdrawal latency was calculated for each animal.

Baseline paw withdrawal thresholds in the Von Frey assay and withdrawal latencies in the Hargreaves assay were measured over 3 sessions (1 baseline session/day; baseline sessions for each assay occurred on alternating days; *Figure 7B*) that were each preceded 30 min earlier by a vehicle (5% DMSO in saline, i.p.) pretreatment. After the final baseline session, rats were counterbalanced into CNO (4 mg/kg) or Vehicle (5% DMSO) treatment groups based on paw withdrawal latencies during the 3$^{rd}$ (final) Hargreaves baseline session. During Von Frey and Hargreaves test sessions, rats were given CNO (4 mg/kg) or vehicle injections (i.p.) 30 min before the start of testing. All Von Frey and Hargreaves procedures occurred under regular white light illumination.

## Anxiety-like behaviors

One day after Hargreaves testing, rats were tested for anxiety-like behaviors in the elevated plus maze (EPM), open field (OF), and light-dark box (LD) on consecutive days (*Figure 7B*). All procedures occurred under indirect, dim illumination (~10 lux). Rats were given CNO (4 mg/kg) or vehicle injections (i.p.) 30 min before the start of each test. The EPM and OF tests were performed as previously described (*Albrechet-Souza et al., 2020*; *Fucich et al., 2020*). Briefly, the EPM consists of two open and two closed arms elevated 50 cm above the floor. Rats were individually placed in the center of the maze facing an open arm and were given 5 min to explore the maze. Time spent in the open and closed arms of the maze was measured. The OF consists of a square arena with a checkerboard patterned floor (4 × 4 squares). Rats were individually placed in one corner of the arena and were given 5 min to explore the arena. Time spent in the periphery and the center of the arena (defined as the 3 × 3 squares in the center of the arena) was measured. The LD box consists of a two compartments; one with black walls and a black floor, and the other with white walls and a white floor. The black compartment was protected from light (dark box) and the white compartment was illuminated (light box; ~ 1000 lux). Rats were able to freely explore both dark and light boxes through an opened door. Rats were individually placed in the dark box and were given 5 min to explore the apparatus. Time spent in dark and light boxes, as well as the latency to enter the light box were measured. EPM, OF, and LD tests were recorded via a camera mounted above the apparatus and videos were scored

by an experimenter blinded to treatment groups. At the end of the experiment, rats were sacrificed and brain sections were analyzed for virus placement.

### *Crhr1* and *iCre* expression in BLA

The purpose of this experiment was to examine *Crhr1* and *iCre* mRNA expression in a brain area outside the CeA. The basolateral amygdala (BLA) was selected for analysis because $CRF_1$ receptors are highly expressed in, and functionally regulate the activity of pyramidal glutamatergic neurons in this area (as opposed to GABAergic neurons in CeA) (*Refojo et al., 2011*; *Rostkowski et al., 2013*). Brain sections containing BLA were processed for *Crhr1* and *iCre* RNAscope and analyzed as described above.

### Cre-<sup>td</sup>Tomato fluorescent expression across the brain

The purpose of this experiment was to examine if the Cre-$^{td}$Tomato transgene is expressed in brain areas outside of the amygdala in a pattern that is consistent with known $CRF_1$ expression patterns. Male and female (n = 2/sex) $CRF_1$-Cre-$^{td}$Tomato rats were perfused transcardially with PBS and 4% PFA, and brains were post-fixed in 4% PFA overnight. Brains were cryoprotected, snap-frozen, and sectioned with a cryostat. Coronal sections (40 µm) were collected across the rostrocaudal axis, mounted onto microscope slides, and coverslipped with Fluorogel-II containing DAPI (Electron Microscopy Sciences). Brain sections were imaged using a Keyence BZ-X800 fluorescent microscope. For each brain region, at least three sections were analyzed.

### Statistical analyses

Electrophysiology data on frequency and amplitude of spontaneous inhibitory postsynaptic currents (sIPSCs) were analyzed and manually confirmed using a semi-automated threshold-based detection software (Mini Analysis, Synaptosoft Inc, Decatur, GA). Cell-attached firing discharge data were analyzed and manually confirmed using a semi-automated threshold-based detection software (Clampfit 10.6, Molecular Devices). Electrophysiological characteristics were determined from baseline and experimental drug application containing a minimum of 65 events each. Event data were represented as mean ± SEM or mean % change from baseline ± SEM and analyzed for independent significance using a one-sample t-test, compared by paired or unpaired t-test where appropriate. Data analysis and visualization were completed using Prism 7.0 (GraphPad, San Diego, CA). Behavioral data were analyzed using multifactorial repeated measures ANOVAs (for Von Frey and Hargreaves tests) or t-tests (for anxiety tests). For RM ANOVAs, between-subjects factors include sex and treatment, and the within-subjects repeated measure was test session (i.e. baseline vs. test). Data from the active and control virus groups were analyzed separately (i.e. the control virus group was treated as a replication of the experiment; *Weera et al., 2021*). Data from experiments that only have two groups (e.g. c-Fos immunohistochemistry) were analyzed using t-tests. Data were analyzed using the Statistical Package for Social Sciences (Version 25, IBM Corporation, Armonk, NY). Statistical significance was set at $p < 0.05$.

## Acknowledgements

This work was supported by National Institutes of Health Grants AA023305 (NWG), AA026022 (to NWG and MAH), AA023002 (MAH), AA027145 (MMW), and AA007577 (institutional NRSA training grant that supported MMW). This work was also supported in part by a Merit Review Award #I01 BX003451 (to NWG) from the United States Department of Veterans Affairs, Biomedical Laboratory Research and Development Service.

# Additional information

## Funding

| Funder | Grant reference number | Author |
|---|---|---|
| National Institute on Alcohol Abuse and Alcoholism | R01 AA023305 | Nicholas W Gilpin |
| National Institute on Alcohol Abuse and Alcoholism | R21 AA026022 | Melissa Herman Nicholas W Gilpin |
| National Institute on Alcohol Abuse and Alcoholism | R00 AA023002 | Melissa Herman |
| National Institute on Alcohol Abuse and Alcoholism | National Research Service Award AA027145 | Marcus M Weera |
| National Institute on Alcohol Abuse and Alcoholism | Institutional Training Grant AA007577 | Marcus M Weera |
| United States Department of Veterans Affairs | Merit Award #I01 BX003451 | Nicholas W Gilpin |

The funders had no role in study design, data collection and interpretation, or the decision to submit the work for publication.

## Author contributions

Marcus M Weera, Conceptualization, Data curation, Formal analysis, Funding acquisition, Investigation, Methodology, Project administration, Writing – original draft, Writing – review and editing; Abigail E Agoglia, Data curation, Formal analysis, Investigation, Writing – original draft; Eliza Douglass, Rosetta S Shackett, Data curation, Investigation; Zhiying Jiang, Shivakumar Rajamanickam, Data curation, Formal analysis, Investigation; Melissa A Herman, Conceptualization, Data curation, Formal analysis, Funding acquisition, Investigation, Methodology, Project administration, Resources, Supervision, Validation, Visualization, Writing – original draft, Writing – review and editing; Nicholas J Justice, Conceptualization, Funding acquisition, Investigation, Project administration, Resources, Supervision, Validation, Writing – review and editing; Nicholas W Gilpin, Conceptualization, Funding acquisition, Project administration, Resources, Supervision, Writing – review and editing

## Author ORCIDs

Marcus M Weera http://orcid.org/0000-0002-2451-0350
Nicholas W Gilpin http://orcid.org/0000-0001-8901-8917

## Ethics

All animal procedures were conducted in accordance with recommendations in the Guide for the Care and Use of Laboratory Animals of the National Institutes of Health, and were approved by the Institutional Animal Care and Use Committee of the respective institutions at which procedures occurred (Louisiana State University Health Sciences Center, University of North Carolina - Chapel Hill, University of Texas Health Sciences Center). (LSUHSC IACUC Protocol #3749; UNC IACUC Protocol #19-190; UTHSC IACUC Protocol #21-075).

## Decision letter and Author response

Decision letter https://doi.org/10.7554/eLife.67822.sa1
Author response https://doi.org/10.7554/eLife.67822.sa2

# Additional files

## Supplementary files

• Transparent reporting form

- Source data 1. Slice electrophysiology data.
- Source data 2. Behavioral data.

**Data availability**
All data generated during this study are included in the manuscript.

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
