## [Editor Report]

This manuscript details the generation and characterization of a new transgenic rat line presenting an exciting new tool to the field.

---

## [Decision Letter]

**Decision letter after peer review:**

Thank you for submitting your article "Cellular, physiological, and behavioral validation of a CRFR1:Cre-td Tomato rat for use in basic neuroscience research" for consideration by *eLife*. Your article has been reviewed by 4 peer reviewers, and the evaluation has been overseen by a Reviewing Editor and Kate Wassum as the Senior Editor. The following individuals involved in review of your submission have agreed to reveal their identity: James P Herman (Reviewer #2); Eric Zorrilla (Reviewer #3).

The reviewers have discussed their reviews with one another, and the Reviewing Editor has drafted this to help you prepare a revised submission. (We suggest resubmitting as a Tools and Resources paper.)

Essential revisions:

1) As previous transgenic rat lines have shown some issues with cell specific fidelity, and since this is the initial characterization paper of the rat line, it was felt that a broader characterization of the expression in different cell types and in different brain regions was necessary. For example, the Pomerenze Crh-Cre rat was found to only express in GABAergic but not glutamatergic neurons, and so for the field to be able to understand the breadth of utility of this model and the fidelity across cell types, more characterization is required. This does not need to involve in depth electrophysiology as was done in the CeA, but could be done by mapping the expression of iCre/tdTomato in brain regions beyond the CeA, such as prefrontal cortex, hypothalamic nuclei and the BNST, where CRHR1 neurons are widely expressed. Ideally this would be compared to ISH quantification as is done in the CeA but if this is unfeasible then comparing to established in situ hybridization maps of expression of CRHR1 would still provide an idea of the fidelity with endogenous CRFR1 expression. It was felt necessary that at least in one other brain region, preferably one where CRHR1 is expressed in glutamatergic neurons, a side by side comparison of tdTomato and CRFR1 would be necessary to be sure it is not restricted to discrete cell types. It would also be good to establish if expression of tdTomato is seen in microglia or astrocytes to know if this model recapitulates the cell type specificity of CRFR1 expression.

Concerns that require addressing but can be dealt with via reanalysis of existing data or modifications within the manuscript.

1) The authors contextualize the necessity of developing a rat model of the CRFR1-Cre line because they state that this will allow more for the investigation of more complex behaviors then mice engage in. While this point may have some validity, it was felt that this statement and justification also required more validity. Please expand this rationale including examples of complex behavioral assays that can only be performed in rats and not mice.

2) The manuscript does not discuss possible genetic strain of origin issues. The rat was generated using a BAC that involves DNA from a different strain (F344) than that bred into (Wistar), so potential strain of origin differences in regulatory or enhancer elements for the Crhr1 gene might co-segregate in early generations with the transgene. Similar considerations are true with respect to linked alleles if the strain of the gamete donors, which was not specified, differed from Wistar. Such issues would become of lesser concern with continued Wistar breeding but are not mentioned.

Additionally, the location, copy #, fidelity or potential mosaicism of BAC transgene insertions after germline transmission was not mentioned. It is recommended that analysis of F1 or F2 offspring to establish location, copy #, fidelity and mosaicism of transgene insertion in order to disseminate the most desirable pedigree from this line (e.g., PMID: 17882484, 29703985, 30354251). If already done, please add relevant findings.

3) For the electrophysiology studies, adding the resting potential on the rep traces in Figure 5 and describing if Vm was adjusted for these experiments would provide clarity and is requested. Also, it could be very useful to compare intrinsic membrane results with published work on these cells to establish the normality of these measures in this line and see if the transgene expression impacted the physiological properties of these cells. It was also felt that the many of the traces used in the figures didn't seem representative and these should be modified to include genuine representative traces.

4) The reviewers all appreciate the authors inclusion of both sexes, but there are some issues here that need to be addressed. First, it is likely that this study was not appropriately powered for identifying sex differences. This is recognized in some experiments, such as the anxiety behavior experiments, but it was agreed that the same level of preliminary interpretation should be exercised with the Cre fidelity data and the impacts of CRF in firing rates. However, sex differences in baseline firing rates are important and it would benefit the manuscript to discuss this in greater depth.

5) The statistics on the behavior (Figure 7) should be checked as it was felt that repeated measures ANOVA would be more appropriate for these then t-tests.

6) Consistent with IUPHAR consensus (p.24, PMID: 12615952), the receptor protein should be referred to as CRF(subscript)1. Per both IUPHAR and RGD (RGD ID: 61276) nomenclature, the gene should be referred to as Crhr1.

7) As the DREADD studies performed did not include an approach that inhibited these cells, the discussion should clarify that it remains to be determined if this approach will work for inhibition of cell activity using actuators like DREADDs.

8) Some aspects of histochemical concordance were not provided or explored. For example, the proportion of Cre-expressing neurons that expressed CRF1 mRNA was not reported to confirm specificity (it appears to be even higher which would be good)? Cellular co-expression of Tomato with CRF1 mRNA also was not explicitly quantified.

To address this, it is recommended to: (a) justify the current binary thresholds for + vs. – expression similar to PMID: 31451604, (b) analyze whether there is a semi-continuous, rather than only binary, relation between puncta expression of CRF1 vs. iCre, (c) clarify whether there are statistically significant sex differences in binary or continuous concordance (CRF1-iCre), and (d) discuss concordance of Tomato with CRF1 mRNA.

9) While anxiety-like behavior was viewed as an appropriate behavioral readout for validation of the rats, it was felt that the consensus around the role of the CeA in pain was less widely accepted and had more nuance. As such, it is requested that authors to cite the work on CeA CRF and analgesia and acknowledge that their chemogenetic manipulation seem to mimic injury-induced hyperalgesic states but not stress-induced analgesic states. The authors can use this opportunity to highlight that the transgenic rat would be an excellent tool for future studies aimed at studying stress-induced analgesia.

10. If you have not already done so, please attend to the important information below "Information to help as you revise your submission" including inclusions of a key resource table and statistical reporting.

[Editors' note: further revisions were suggested prior to acceptance, as described below.]

Thank you for submitting your article "Generation of a CRF_1_-Cre transgenic rat and the role of central amygdala CRF_1_ receptors in nociception and anxiety-like behavior" for consideration by *eLife*. Your article has been reviewed by 2 peer reviewers, and the evaluation has been overseen by a Reviewing Editor and Kate Wassum as the Senior Editor. The reviewers have opted to remain anonymous.

Essential revisions:

The authors did an adequate job in answering many of the reviewer’s concerns, however, it does seem that the author response fell short in answering two of the reviewers' concerns.

Specifically, there was a consensus amongst reviewers that analysis of the penetrance and specificity of transgene across several regions should be included in this first demonstration of a new transgenic tool. While the inclusion of the BLA is an improvement, it still does not survey several regions. Rather the authors indicate that a full analysis of the specificity and penetrance of the transgene across brain regions will be reserved for a follow-up manuscript. While this may be, the reviewers agreed that it is paramount to this first publication to include more than two regions. From discussion, it was agreed amongst the reviewers that the authors should include characterization of the following brain regions: BNST, Hippocampus, VTA, LC, RN, PAG, PVN, and at least one cortical region.

The second concern that does not seem to be sufficiently addressed in the revision is the need to perform experiments to determine where the BAC incorporated in the genome in the CRF1-Cre rat. This is particularly important when considering fluorescent reporter expression. The reviewers appreciated the additional discussion of potential caveats but agreed that additional experiments and analysis to validate this component of a new tool are needed. The issues that have arisen from other BAC mice and rats are believed to be a result of less than rigorous validation upon their initial introduction. As such, it seems incumbent on the authors to be as rigorous as possible in this first publication in order to avoid these recurrent issues with this particular rat tool.

---

## [Author Response]

Essential revisions:1) As previous transgenic rat lines have shown some issues with cell specific fidelity, and since this is the initial characterization paper of the rat line, it was felt that a broader characterization of the expression in different cell types and in different brain regions was necessary. For example, the Pomerenze Crh-Cre rat was found to only express in GABAergic but not glutamatergic neurons, and so for the field to be able to understand the breadth of utility of this model and the fidelity across cell types, more characterization is required. This does not need to involve in depth electrophysiology as was done in the CeA, but could be done by mapping the expression of iCre/tdTomato in brain regions beyond the CeA, such as prefrontal cortex, hypothalamic nuclei and the BNST, where CRHR1 neurons are widely expressed. Ideally this would be compared to ISH quantification as is done in the CeA but if this is unfeasible then comparing to established in situ hybridization maps of expression of CRHR1 would still provide an idea of the fidelity with endogenous CRFR1 expression. It was felt necessary that at least in one other brain region, preferably one where CRHR1 is expressed in glutamatergic neurons, a side by side comparison of tdTomato and CRFR1 would be necessary to be sure it is not restricted to discrete cell types. It would also be good to establish if expression of tdTomato is seen in microglia or astrocytes to know if this model recapitulates the cell type specificity of CRFR1 expression.

We now show Crhr1 and iCre co-expression in the basolateral amygdala, a brain area that contains a high density of Crhr1-expressing glutamatergic neurons (page 20, lines 263-271; page 39, lines 692-698). A follow-up manuscript will characterize Crhr1 and iCre co-expression throughout the rat brain.

Concerns that require addressing but can be dealt with via reanalysis of existing data or modifications within the manuscript.1) The authors contextualize the necessity of developing a rat model of the CRFR1-Cre line because they state that this will allow more for the investigation of more complex behaviors then mice engage in. While this point may have some validity, it was felt that this statement and justification also required more validity. Please expand this rationale including examples of complex behavioral assays that can only be performed in rats and not mice.

Please see page 4, lines 70-71, and page 8, lines 143-145.

2) The manuscript does not discuss possible genetic strain of origin issues. The rat was generated using a BAC that involves DNA from a different strain (F344) than that bred into (Wistar), so potential strain of origin differences in regulatory or enhancer elements for the Crhr1 gene might co-segregate in early generations with the transgene. Similar considerations are true with respect to linked alleles if the strain of the gamete donors, which was not specified, differed from Wistar. Such issues would become of lesser concern with continued Wistar breeding but are not mentioned.Additionally, the location, copy #, fidelity or potential mosaicism of BAC transgene insertions after germline transmission was not mentioned. It is recommended that analysis of F1 or F2 offspring to establish location, copy #, fidelity and mosaicism of transgene insertion in order to disseminate the most desirable pedigree from this line (e.g., PMID: 17882484, 29703985, 30354251). If already done, please add relevant findings.

We agree that the insertion site and copy number are important considerations that may influence transgenic expression of reporter genes. We have not yet performed experiments to determine where the BAC incorporated in the genome in the CRF1-Cre rat, however, we have included these as important caveats to consider when analyzing transgenic expression, in the Discussion (pages 21 and 22, lines 295-313).

3) For the electrophysiology studies, adding the resting potential on the rep traces in Figure 5 and describing if Vm was adjusted for these experiments would provide clarity and is requested. Also, it could be very useful to compare intrinsic membrane results with published work on these cells to establish the normality of these measures in this line and see if the transgene expression impacted the physiological properties of these cells. It was also felt that the many of the traces used in the figures didn't seem representative and these should be modified to include genuine representative traces.

The cell-attached recordings of neuronal firing were performed with no holding parameters (0 mV) and no adjustments to Vm were made. The intrinsic properties of CRF_1_^+^ CeA neurons are in line with what has previously been reported in rat CeA neurons (Herman & Roberto, 2016; PMID: 25170988), suggesting that the transgene did not significantly impact the overall health or activity of CRF_1_^+^ CeA neurons (please see page 22, lines 315-317). The figures were edited to include traces that were more representative of the CRF_1_^+^ population in males and females (please see Figure 5).

4) The reviewers all appreciate the authors inclusion of both sexes, but there are some issues here that need to be addressed. First, it is likely that this study was not appropriately powered for identifying sex differences. This is recognized in some experiments, such as the anxiety behavior experiments, but it was agreed that the same level of preliminary interpretation should be exercised with the Cre fidelity data and the impacts of CRF in firing rates. However, sex differences in baseline firing rates are important and it would benefit the manuscript to discuss this in greater depth.

Please see page 23, lines 330-331; page 26, lines 410-413. Given that our studies here were not sufficiently powered to conclusively determine the presence or absence of sex differences, we feel that the sex difference in baseline firing rate of CeA CRF_1_ cells that we observed should not be over-interpreted.

5) The statistics on the behavior (Figure 7) should be checked as it was felt that repeated measures ANOVA would be more appropriate for these then t-tests.

Data from Von Frey and Hargreaves tests were analyzed using repeated measures ANOVA. Data from elevated plus maze, open field, and light-dark tests were analyzed using t-tests because there are 2 between-subjects groups in these experiments. This is now clarified in the manuscript (please see Results, page 18, lines 241, 249 and 255; Statistical Analyses, page 40, lines 710713).

6) Consistent with IUPHAR consensus (p.24, PMID: 12615952), the receptor protein should be referred to as CRF(subscript)1. Per both IUPHAR and RGD (RGD ID: 61276) nomenclature, the gene should be referred to as Crhr1.

This has been edited throughout the manuscript.

7) As the DREADD studies performed did not include an approach that inhibited these cells, the discussion should clarify that it remains to be determined if this approach will work for inhibition of cell activity using actuators like DREADDs.

This has been clarified in the Discussion (please see page 23, line 339; page 25, lines 399-401).

8) Some aspects of histochemical concordance were not provided or explored. For example, the proportion of Cre-expressing neurons that expressed CRF1 mRNA was not reported to confirm specificity (it appears to be even higher which would be good)? Cellular co-expression of Tomato with CRF1 mRNA also was not explicitly quantified.To address this, it is recommended to: (a) justify the current binary thresholds for + vs. – expression similar to PMID: 31451604, (b) analyze whether there is a semi-continuous, rather than only binary, relation between puncta expression of CRF1 vs. iCre, (c) clarify whether there are statistically significant sex differences in binary or continuous concordance (CRF1-iCre), and (d) discuss concordance of Tomato with CRF1 mRNA.

Please see edits on page 35, line 590 for information on our RNAscope quantification thresholding. Beyond counting cells as being positively or negatively labeled with an RNAscope probe, we did not quantify the number of puncta within each cell as this re-analysis would be difficult without specialized software. We feel that this level of analysis is beyond the scope of the current manuscript. Regarding potential sex differences, we did analyze RNAscope data with sex as a factor, using binary thresholds (page 9, lines 161-153). Regarding concordance between ^td^Tomato and Crhr1 mRNA expression, we probed for iCre and not ^td^Tomato because iCre and ^td^Tomato are expressed as a single polypeptide (please see edits on page 9, lines 154-156). Therefore, there is no reason to suspect any difference between iCre and ^td^Tomato expression, and we probed for iCre because it is more relevant for functional studies.

9) While anxiety-like behavior was viewed as an appropriate behavioral readout for validation of the rats, it was felt that the consensus around the role of the CeA in pain was less widely accepted and had more nuance. As such, it is requested that authors to cite the work on CeA CRF and analgesia and acknowledge that their chemogenetic manipulation seem to mimic injury-induced hyperalgesic states but not stress-induced analgesic states. The authors can use this opportunity to highlight that the transgenic rat would be an excellent tool for future studies aimed at studying stress-induced analgesia.

Please see Discussion, page 24, lines 355-365.

10. If you have not already done so, please attend to the important information below "Information to help as you revise your submission" including inclusions of a key resource table and statistical reporting.

A key resources table is now included at the beginning of the Materials and methods section (pages 28-29). Statistical reporting has been checked per *eLife* instructions.

[Editors' note: further revisions were suggested prior to acceptance, as described below.]

The reviewers have discussed their reviews with one another, and the Reviewing Editor has drafted this to help you prepare a revised submission.Essential revisions:The authors did an adequate job in answering many of the reviewer’s concerns, however, it does seem that the author response fell short in answering two of the reviewers' concerns.Specifically, there was a consensus amongst reviewers that analysis of the penetrance and specificity of transgene across several regions should be included in this first demonstration of a new transgenic tool. While the inclusion of the BLA is an improvement, it still does not survey several regions. Rather the authors indicate that a full analysis of the specificity and penetrance of the transgene across brain regions will be reserved for a follow-up manuscript. While this may be, the reviewers agreed that it is paramount to this first publication to include more than two regions. From discussion, it was agreed amongst the reviewers that the authors should include characterization of the following brain regions: BNST, Hippocampus, VTA, LC, RN, PAG, PVN, and at least one cortical region.

We agree with the reviewers, and we have performed additional experiments to image and quantify Cre-tdTomato-expressing cells in multiple areas across the CRF1-Cre-tdTomato rat brain. We now show that Cre-tdTomato-expressing cells are found in brain areas known to contain CRF1 cells, including the prelimbic and infralimbic cortex, piriform cortex, bed nucleus of stria terminalis, hippocampus, paraventricular nucleus of hypothalamus, lateral hypothalamus, paraventricular thalamus, ventral tegmental area, dorsal raphe, and locus coeruleus (Pg. 19-22, Figure 9). We compared the density of Cre-tdTomato cells in these areas to expression levels of Crhr1 mRNA in wild type rats (Van Pett et al., 2001), and density of GFP cells in CRF_1_-GFP mice (Justice et al.,2008), and found that they are similar (Pg. 22, Table 1).

The second concern that does not seem to be sufficiently addressed in the revision is the need to perform experiments to determine where the BAC incorporated in the genome in the CRF1-Cre rat. This is particularly important when considering fluorescent reporter expression. The reviewers appreciated the additional discussion of potential caveats but agreed that additional experiments and analysis to validate this component of a new tool are needed. The issues that have arisen from other BAC mice and rats are believed to be a result of less than rigorous validation upon their initial introduction. As such, it seems incumbent on the authors to be as rigorous as possible in this first publication in order to avoid these recurrent issues with this particular rat tool.

We understand this concern and we performed additional experiments to try to identify the genomic insertion site of the BAC. Unfortunately, these experiments were not successful, and we were not able to conclusively determine the insertion site. We explored a commercial sequencing option, but we currently do not have the resources for this option. Given that the expression pattern of the fluorescent reporter in CRF1-Cre-tdTomato rats is consistent with the known expression pattern of CRF1 cells throughout the brain, we are confident that genomic incorporation of the transgene produced expected Cre-tdTomato expression patterns. In addition, the behavioral and physiological phenotypes of CRF1-Cre-tdTomato rats reported in this manuscript closely resemble those seen in wild-type Wistar rats, suggesting that BAC insertion into the genome did not produce observable anomalous phenotypes (added to Discussion on Pg. 24).